# Photothermal modulated dielectric elastomer actuator for resilient soft robots

Matthew Wei Ming Tan[1], Hyunwoo Bark[1], Gurunathan Thangavel[1], Xuefei Gong[1] & Pooi See Lee [ID] [1] ✉

Soft robots need to be resilient to extend their operation under unpredictable environments. While utilizing elastomers that are tough and healable is promising to achieve this, mechanical enhancements often lead to higher stiffness that deteriorates actuation strains. This work introduces liquid metal nanoparticles into carboxyl polyurethane elastomer to sensitize a dielectric elastomer actuator (DEA) with responsiveness to electric fields and NIR light. The nanocomposite can be healed under NIR illumination to retain high toughness (55 MJ m$^{-3}$) and can be recycled at lower temperatures and shorter durations due to nanoparticle-elastomer interactions that minimize energy barriers. During co-stimulation, photothermal effects modulate the elastomer moduli to lower driving electric fields of DEAs. Bilayer configurations display synergistic actuation under co-stimulation to improve energy densities, and enable a DEA crawler to achieve longer strides. This work paves the way for a generation of soft robots that achieves both resilience and high actuation performance.

Soft robots are preferred to be operated under unstructured and unpredictable conditions due their adaptable characteristics[1,2]. However, soft materials utilized tend to be mechanically weak and are prone to damage from large tensile loads and sharp objects. After damage, the material loses its functionality, marking the end of life of the soft robot. This highlights the need for resilient soft robots with the ability to resist and recover from damages, prolonging their operational lifetime[3,4]. Drawing inspiration from their biological counterparts, damage resistance can be achieved through utilizing elastomers with mechanical properties of high strength and toughness[5,6]. When damage occurs, materials imparted with self-healing and recyclability enable soft devices to be easily recovered for continued functionality[4,7]. This recovery process should be of low-energy to be sustainable and economically favorable for greater adoption[8].

Soft actuators are a major class of soft devices that have contributed greatly to the rise of soft robots. These actuators are composed of materials that respond to various stimuli, including light[9], humidity[10], electric/magnetic fields[5,11], and heat[12], to achieve mechanical deformations. Among these soft actuators, dielectric elastomer actuators (DEAs) have achieved large actuation strains, rapid speeds and large energy densities, showing promises for applications including soft robotics, haptics and wearables[13–15]. DEAs are composed of an elastomer sandwiched between two compliant electrodes, at which it operates with the application of an electric field that generates a Maxwell pressure to induce deformation. While previous works have made DEAs healable for prolonged lifetime[16,17], imparting high strength and toughness to these actuators remains to be achieved. This can be attributed to the modulus increase that is often accompanied by mechanical property enhancements that impede actuation strains or raise driving electric field requirements[13]. Ensuring that high actuation performances are achieved while in pursuit of resilient soft DEAs are crucial to increase the practical adoption of these strategies. To achieve this, this work introduces the concept of co-stimulation.

Co-stimulation of actuators has been explored at which two stimuli are applied concurrently to achieve a particular function[18,19]. Such co-activation is analogous to athletes warming up their muscles before

[1]School of Materials Science and Engineering, Nanyang Technological University, 50 Nanyang Avenue, Singapore 639798, Singapore. ✉e-mail: pslee@ntu.edu.sg

an exercise. While muscles are activated by neural electrical signals, the addition of a co-stimuli to raise the body temperature results in increased blood flow to the muscles, sensitivity of nerve receptors and speed of nervous impulses for improved muscle performance[20,21]. Melvin et al. relied on co-stimulation of light and electric field to drive a polypyrrole-TiO$_2$ electrochemical actuator in an aqueous solution[18]. Light was used to generate electron-hole pairs in the TiO$_2$ region. At the same time, the applied electric field directs the charges to the opposing ends of the actuator to induce redox reactions for actuation. However, when activated by a single stimuli the actuator exhibits no response, limiting the device's operational conditions. Co-stimulation for a reconfigurable actuator has been explored through photothermal heating and magnetic fields that enable softening of shape memory polymers and setting of temporary shapes, respectively[19]. Yet, heating and magnetic deformations are both dynamically reliant on the positioning of both external energy sources, which further complicates the entire design[19,22]. These concurrent dynamic changes create greater challenges in predicting and modeling deformations[22]. Very often, the two stimuli applied perform independent roles with minimal synergy. To fully realize the potential of co-stimulation effects, the design and selection control of co-stimuli remains critical to achieving direct control and synergistic responses that amplify or augment actuation capabilities[23].

In pursuit of resilient soft robots with high actuation performances, co-stimulation of NIR light and electric fields are applied to achieve tough, self-healable and recyclable DEAs. Polyurethane elastomer is functionalized with carboxyl groups that provide a strong and dynamic physical crosslinked network through hydrogen bonding. Liquid metal nanoparticles (LMNPs) are introduced to the matrix due to their unique ability to increase dielectric constants and concurrently impart photothermal capabilities to the elastomer, improving responses towards both electric fields and NIR light. Photothermal effects can be used to promote self-healing of the liquid metal nanocomposites owing to supramolecular hydrogen bonding. Compared to the pristine polyurethane, the resultant liquid metal nanocomposites (PULMX, X indicating the wt% of LMNPs) can be recycled at lower temperatures and shorter times of 100 °C and 15 min, respectively, to retain 90% of its stretchability. The reduced energy requirement for recycling nanocomposites can be attributed to the disruption of hydrogen bonds between polymer chains with LMNPs, as revealed from structure-property characterizations. Through which, we report a recyclable DEA that achieves 105% area strain and maintains 57.1% area strain after recovery. Unlike liquid metal microparticles that are conductive under mechanical strains and rely on thermal effects to achieve electrothermal and photothermal actuation[9,24,25], LMNPs remain insulative[26,27], preventing dielectric breakdown to allow Maxwell pressures and thermal effects to be used simultaneously for co-stimulation. In addition, nanoscale liquid metal particles have higher photothermal conversion efficiencies[26]. More importantly, when cooled, the elastomer can retain its mechanical properties, ensuring damage resistance to external threats. When co-stimulation is applied to pre-strain free DEAs (Supplementary Fig. 1), the driving electric field to achieve 50% area strains is lowered by 40% due to the modulus reduction from photothermal heating. Dielectric minimum energy structures (DEMES) are further explored (Supplementary Fig. 2), at which multi-stimuli response to NIR light and electric field can be achieved based on differential thermal expansion and Maxwell pressures, respectively. Under co-stimulation, a synergistic effect is found between Maxwell pressure and thermal expansion due to photothermal softening effects. When applied to a soft crawler driven by DEAs, photothermal effects enable longer strides and increase the flattening of its body for enhanced shape adaptability to move across narrow spaces. Through co-stimulation, resilient elastomers can be applied to soft robots for longer operational lifetimes with high actuation capabilities.

## Results

### Synthesis and characterization of PULM nanocomposites

A one-step synthesis between the mixture of polytetrahydrofuran (PTMG), 2,2-bis(hydroxymethyl)propionic acid (DMPA) and isophorone diisocyanate (IPDI) was performed to prepare PULM0 (Fig. 1a and Supplementary Fig. 3). The successful synthesis of PULM0 was verified through $^1$H nuclear magnetic resonance and fourier transform infrared spectroscopy (FTIR) (Supplementary Figs. 4 and 5). Bulk liquid metal (LM) underwent probe sonication in dimethylformamide (DMF) to achieve liquid metal nanoparticles (LMNPs) that appeared darkish gray after drying instead of its original reflective silver (Fig. 1b). Size distribution of LMNPs was evaluated through scanning electron microscope (SEM) and dynamic light scattering (DLS) at which particle sizes were revealed to be within hundreds of nanometers (Fig. 1c, d). High compatibility between LM and PULM0 was found as contact angles reduced from 139.6° to 91.2° within 480 min. (Fig. 1e and Supplementary Fig. 6). This can be attributed to hydrogen bond formation between carboxyl functionalities of PULM0 and the native gallium oxide shell surrounding LMNPs (Supplementary Fig. 7)[25,28]. These interactions were further investigated through FTIR analysis of nanocomposites (Supplementary Figs. 8 and 9). Distinct peak shifts towards higher wavenumbers at C = O regions can be observed, indicating that increasing LMNP concentrations lead to further hydrogen bonding between PULM0 and LMNPs, while reducing the hydrogen bonding between PULM0 polymer chains concurrently[28]. Thermal transitions of the nanocomposites were studied through differential scanning calorimetry (DSC) (Fig. 1f and Supplementary Fig. 10) and dynamic mechanical analysis (DMA) (Supplementary Fig. 11). When compared to the thermal transition of bulk LM (Supplementary Fig. 12), a suppression of peak freezing (−5.8 to −63.8, and −48.7 °C) and melting points (11.5 to −41.3, and −30.7 °C) was observed for LMNPs. This is attributed to supercooling effects that arise from the phase separation between solid and liquid phases and the formation of their interface[29]. Owing to the larger molecular weight of PTMG used, PULM0 soft segments formed crystalline structures that were evident from endothermic melting peaks at 21 °C[30,31]. Melting enthalpies decreased from 35.0 J/g to 13.6 J/g with higher LMNPs concentrations, indicating a disruptive role of LMNPs to the PULM0 polymer structure and crystalline nature. This effect was further corroborated with wide angle X-ray scattering (WAXS) measurements where the sharpness and intensity of diffractions peaks found at $2\theta = 20.1°$ were reduced (Fig. 1g). Concurrently, PULM0 displayed microphase separation between hard and soft phases based on a broad scattering peak at $q \sim 0.07$ A$^{-1}$ from small angle X-ray scattering (SAXS) studies (Fig. 1h). Such characteristic microphase separation can be attributed to the soft segment crystallinity and the strong hydrogen bonding between hard segments with carboxyl groups[5,32]. However, the introduction of LMNPs has diminished the scattering peaks due to increased microphase mixing from new hydrogen bonds formed between LMNPs and PULM0 matrix[28,33–35]. While this interaction may be thermodynamic unfavorable due to the stronger hydrogen bonds between polymer chains, the high surface-to-volume ratio of LMNPs allows such interactions to be dynamically preferred[36].

### Mechanical and photomechanical properties

PULM0 achieved high tensile strengths (18.5 MPa), stretchability (2228%), and mechanical toughness (96.5 MJ m$^{-3}$) owing to the abundant hydrogen bonds and chain entanglements that act as physical crosslinkers (Fig. 2a). With the addition of LMNPs at 15 and 33 wt%, the mechanical properties were further enhanced (Fig. 2b). Increase in elastic modulus with LMNPs is compared with theoretical predictions based on the double inclusion model[37], at which a slightly larger experimental modulus is found (Supplementary Fig. 13). This can be attributed to the formation of bound rubber that surrounds LMNPs due to hydrogen bond interactions[38]. At this interphase, the mobility of

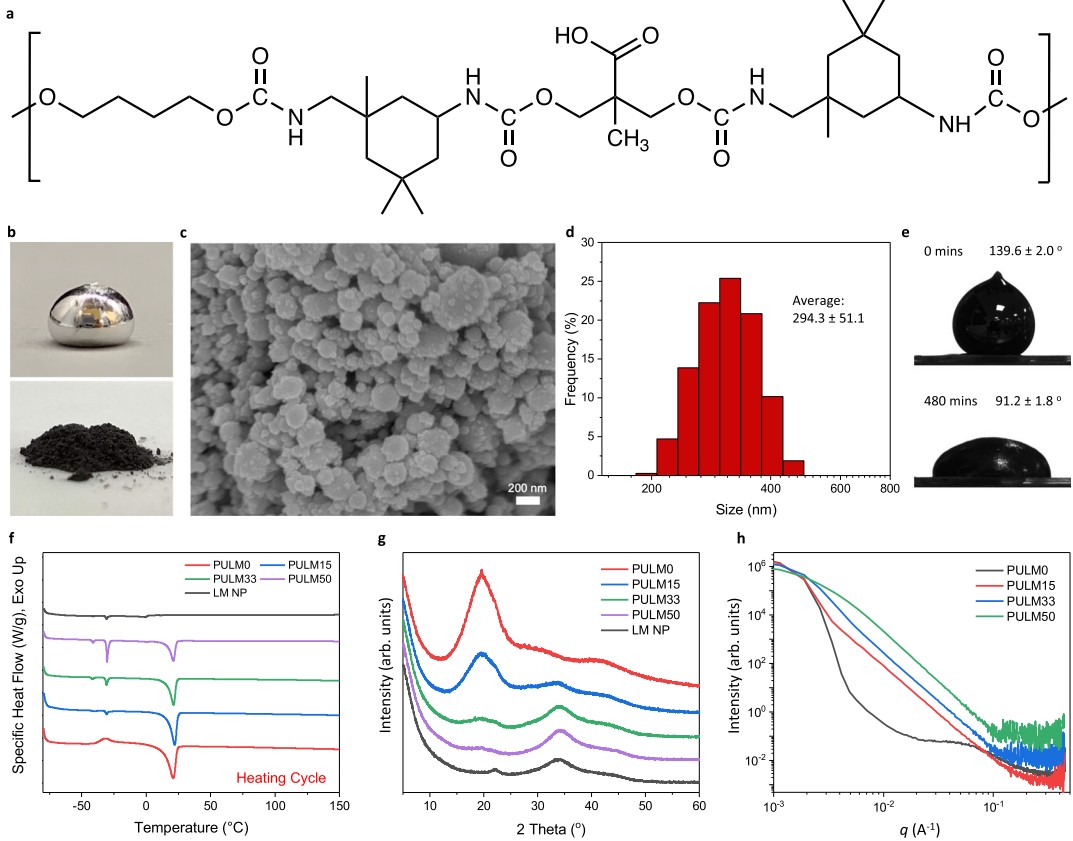

**Fig. 1 | Material characterization of LMNPs and PULM nanocomposites.**
**a** Chemical structure of elastomer matrix PULM0. **b** Digital images of bulk liquid metal with a silver reflective surface and LMNPs that showed a dark gray color after probe sonication and drying. **c** SEM image of obtained LMNPs. **d** Size distribution histogram of LMNPs from DLS measurements **e** Contact angle of bulk liquid metal on PULM0 substrate at 0 min and 480 min. Standard deviation (SD) is obtained from three independent samples. Data is presented in the form of mean values ± SD. **f** DSC heating curves of LMNPs and PULM nanocomposites at a heating rate of 10 °C/min under a nitrogen atmosphere. **g** WAXS comparisons of LMNPs and PULM nanocomposites **h** SAXS comparison between PULM nanocomposites.

polymer chains is restricted, resulting in stiffening effects[39]. Also, the increase in modulus is related to stress effects from gallium oxide shell and particle-elastomer interactions that are elevated as the inclusion diameter is reduced to hundreds of nanometers while the thickness of the gallium oxide shell remains at a few nanometers[27,37]. Improvements to mechanical toughness can be related to effective load transfer from the polymer chains to LMNPs owing to interfacial interactions[40]. At higher amount of LMNPs (50 wt%), the mechanical toughness dropped as the agglomeration of LMNPs increases and results in lowering the strain at break. Energy dissipation characteristics of the nanocomposites were evaluated through cyclic stress-strain tests (Supplementary Fig. 14). Hysteresis losses from PULM0 can be attributed to the rupture of hydrogen bonds and internal friction between polymer chains during stretching[5,34]. For the nanocomposites, internal frictions between polymer chains became pronounced due to the disrupted carboxyl hydrogen bonds. With higher interfacial friction between LMNPs and polymer chains, hysteresis losses increased with LMNP content at strain limits of 100, 500, and 1000% (Fig. 2c). At increasing strain limits, energy dissipation effects are more pronounced as stronger carboxyl hydrogen bonds are being ruptured[5]. To determine the recoverability of the elastomers, elastomers were left for a rest period of 180 min after cycling at similar strain limits (Fig. 2d and Supplementary Fig. 15). At medium strains of 100%, PULM0, PULM15 and PULM33 maintained high recovery ratios while PULM50 was unable to recover its performance due to the disruption imposed by LMNPs against the reformation of broken hydrogen bonds between polymer chains. Deeper insights into the role of LMNPs on the dynamic properties of the elastomeric network were evaluated based on the

Maxwell model, through stress relaxation tests at different temperatures (Supplementary Fig. 16). Relaxation times (time taken to reach 37% of the initial relaxation modulus) reduced with higher LMNPs content and showed a temperature dependence following Arrhenius equation (Fig. 2e and (Supplementary Fig. 17)), at which relaxation activation energies ($E_{a,r}$) were extracted from the slope of the linear fit[41,42]. Activation energies were lowered from 114.2 to 52.3 kJ mol$^{-1}$ with increasing LMNPs (Supplementary Fig. 18), indicating that topological rearrangements can be realized at lower energy barriers and further highlights the role of LMNPs in disrupting strong hydrogen bonds between polymer chains.

LMNPs provide favorable photothermal properties due to optically induced resonance[43,44], and NIR light could serve as a contactless trigger to tune the modulus of the nanocomposites on-demand. Based on UV−Vis spectrophotometry measurements, LMNPs addition led to a strong absorption over a broad spectral band from visible to NIR regions compared to PULM0 which showed minimal absorption (Supplementary Fig. 19). To ensure that photothermal effects did not cause thermal degradation during actuation, upper bound temperatures were determine through thermogravimetric analysis at which less than 1% weight change was observed up to 190 °C (Supplementary Fig. 20). Subsequently, photothermal experiments performed on the nanocomposite films showed significant photothermal conversion with the introduction of LMNPs, saturating at 15 wt% (Supplementary Figs. 21 and 22). The saturation of photothermal conversion can be attributed to the higher tendency of aggregate formation at larger concentrations that minimizes the effective optical resonance behavior[45]. Within 10 s under 0.8 W cm$^{-2}$, the temperature rose by

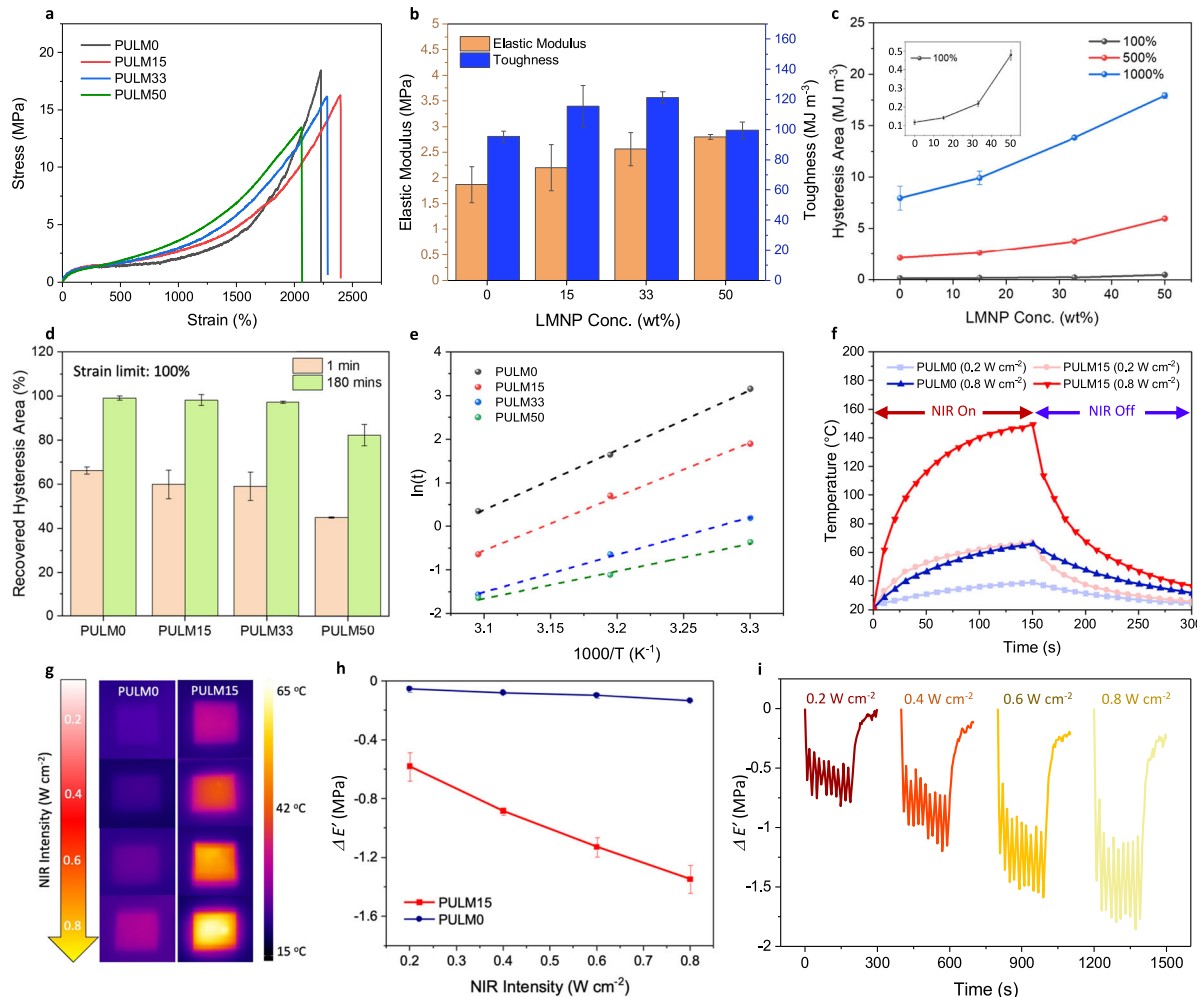

**Fig. 2 | Mechanical, photothermal and photomechanical properties of PULM nanocomposites. a** Tensile stress strain curve of PULM nanocomposites under a strain rate of 100 mm min$^{-1}$. **b** Elastic modulus and mechanical toughness of PULM nanocomposites. **c** Hysteresis area when PULM nanocomposites are pulled to 100, 500, and 1000% strain after the first cycle of cyclic stress–strain measurements. Inset shows the hysteresis area at 100% strain. **d** Recovered hysteresis area of PULM nanocomposite cyclic stress–strain curves after a 1 min and 180 min delay at 100% strain. **e** Arrhenius fittings (dashed lines) of ln(t) versus scaled inverse temperature. t refers to the relaxation time. Slope of Arrhenius fittings represents the relaxation activation energies. **f** Temperature change profile of PULM0 and PULM15. NIR light was turned on for 150 s at 0.2 W cm$^{-2}$ or 0.8 W cm$^{-2}$ and subsequently turned off for films to cool down. **g** IR images of PULM0 and PULM15 illuminated at different NIR light intensity for 150 s. **h** Change in storage modulus ($\Delta E'$) of PULM0 and PULM15 after 10 s of NIR light illumination as a function of NIR light intensity. **i** Change in storage modulus ($\Delta E'$) of PULM15 as a function of time after turning on and off the NIR light for 10 cycles at a frequency of 0.05 Hz followed by cooling for 100 s. All error bars are the standard deviation of three independent samples.

41.3 °C in PUMLP15 while PULM0 showed a lower temperature increase of 6.8 °C (Fig. 2f). Owing to the higher thermal conductivity of PULM15 (Supplementary Fig. 23), elastomers can be cooled at a faster rate. This was evident as PULM0 and PULM15 were heated to similar temperatures (~66 °C) after which the NIR light removal led to temperature reductions of 36 °C and 41 °C, respectively, within 150 s. Photomechanical properties of the nanocomposites were evaluated based on changes in storage modulus upon NIR light illumination. The amount of heat generated can be modulated through different NIR light intensities (Fig. 2g). At higher generated temperatures, the hydrogen bond equilibrium shifted towards the dissociated state, effecting changes in the mechanical properties[46]. Following this principle, upon photothermal generation, PULM15 displayed a reduction of its storage modulus by 1.43 MPa under 0.8 W cm$^{-2}$ respectively (Fig. 2h). Minimal changes were observed for PULM0 with the absence of LMNPs. More importantly, reversible modulation of the storage modulus occurred as the hydrogen bonds reformed upon cooling[47]. This phenomenon is found when the NIR light was alternately turned on and off for 10 cycles at NIR light intensity of 0.2 to 0.8 W cm$^{-2}$

(Fig. 2i). The initial storage modulus can be completely recovered even after extending the repeated stimulation to 100 cycles at 0.8 W cm$^{-2}$ (Supplementary Fig. 24). Compared to other approaches such as plasticizers and molecular designs that reduce the modulus at similar ranges, their changes are permanent or irreversible[48]. In contrast, the use of photothermal effects allows the recovery of its favorable mechanical properties after softening, enabling the elastomer to have greater resistance against external damages.

## Self-healing and recyclability
Based on the dynamic nature of the physical crosslinking network, PULM15 exhibits self-healing and recyclability. Owing to the minimal polymer chain mobility, a mechanical toughness of ~1.4 MJ m$^{-3}$ was recovered at room temperature after 30 min. To improve the recovery of mechanical properties, NIR light can be used to accelerate the thermal diffusion of polymer chains for greater reassociation of bonds across the broken interface. The thermal enhanced self-healing of PULM15 was further evaluated from the recovered mechanical toughness after tensile tests of dumb-bell-shaped samples under

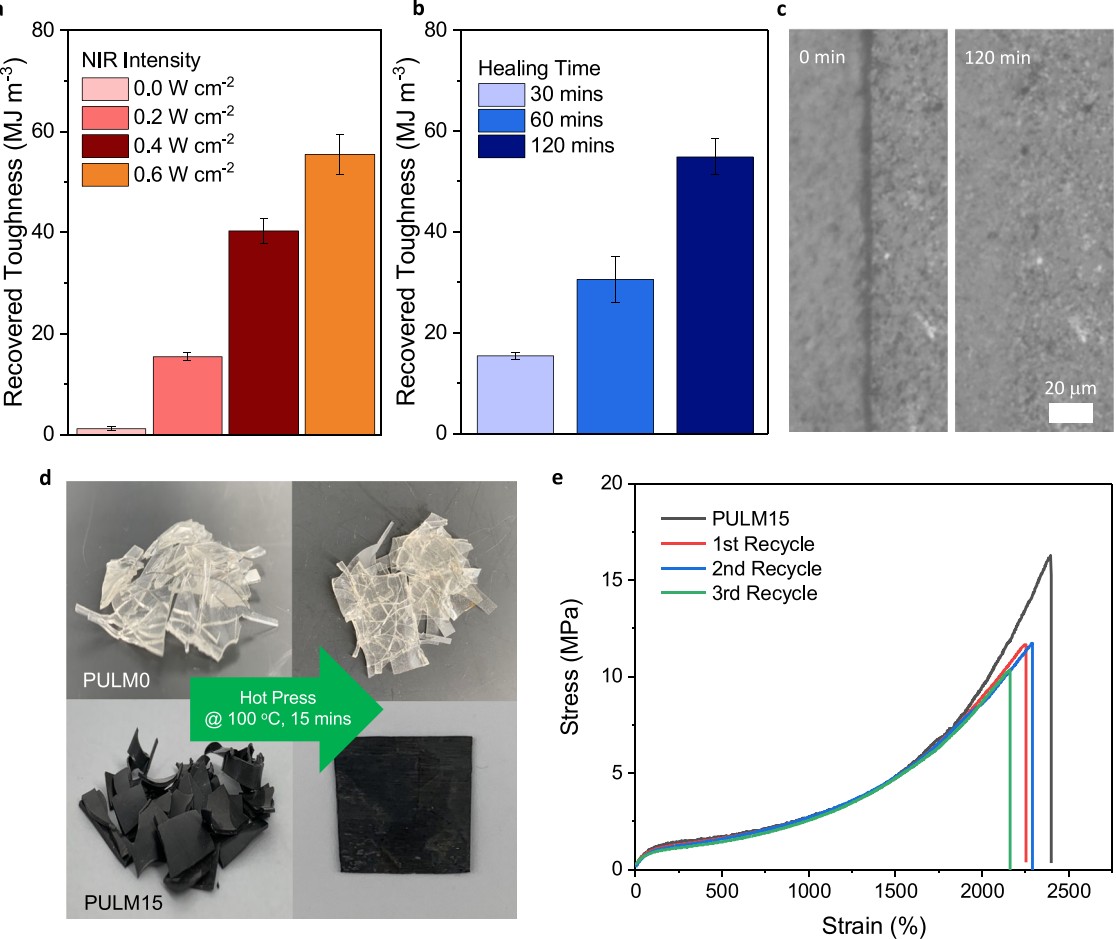

**Fig. 3 | Self-healing and recyclability of PULM15 nanocomposite. a** Recovered toughness of PULM15 at different NIR light intensity for 30 min. **b** Recovered toughness of PULM15 at different healing times under NIR light of 0.2 W cm⁻². **c** Optical microscope images of PULM15 damaged (left, 0 min) and healed after 120 min under NIR light of 0.2 W cm⁻². **d** Digital images of PULM0 and PULM15 waste material before and after hot-pressing for 100 °C for 15 min. PULM0 failed to be recycled while PULM15 was successfully recycled into a film. **e** Tensile stress strain curve of PULM15 and recycled PULM15 (three cutting-recycling cycle) under a strain rate of 100 mm min⁻¹. All error bars are the standard deviation of three independent samples.

varied NIR intensities and durations (Supplementary Fig. 25). By increasing the NIR light intensity from 0.2 to 0.6 W cm⁻², mechanical toughness could be recovered from 18 to 55 MJ m⁻³ within 30 min (Fig. 3a). The improvement is attributed to better interpenetration and diffusion of polymer chains across the broken interface at higher temperatures generated. Being dependent on chain mobilities, increasing the healing time to 120 min allows similar mechanical toughness to be achieved at low NIR light intensity of 0.2 W cm⁻² (Fig. 3b). Complete healing at the surface of the cut interface can also be observed from optical images (Fig. 3c). However, it remains challenging to flawlessly align broken interfaces for optimal recovery[49], particularly with thermal expansion occurring from photothermal effects.

Recyclability was achieved as broken pieces of PULM15 were hot-pressed at 100 °C for 15 min (Fig. 3d). Similar treatment performed on PULM0 did not result in the reformation of broken segments. Instead, recycling of PULM0 could only occur either at extended time periods of 100 °C for 120 min or at higher temperatures of 160 °C for 15 min (Supplementary Fig. 26). When compared to other recyclable elastomers with high tensile strength (>10 MPa), LMNPs enable recycling of the elastomer at lower temperatures and shorter times (Supplementary Table 1). This can be attributed to the role of LMNPs in lowering energy barriers for topological rearrangements through disrupting strong hydrogen bonds between carboxyl functionalities of polymer chains. In addition, the higher thermal conductivity of PULM15 may

allow effective heat transfer to the elastomer matrix for topological rearrangements at short times. Tensile tests of recycled PULM15 retained similar mechanical performance as its original even after three cutting and recycling cycles (Fig. 3e), highlighting the long-term recovery of the elastomer.

## Resilient DEAs

Polar carboxyl groups present in PULM0 provide dipole polarization effects that increase dielectric constant for larger Maxwell pressure, and drive higher actuation[5,50]. With LMNPs, additional Maxwell–Wagner interfacial polarization effect enhances the dielectric constant from 7.6 to 17.3 at 1 kHz (Fig. 4a). Insulative gallium oxide shell further suppresses leakage currents, leading to low dielectric losses (<0.005) and the ability to withstand high driving voltages in DEAs (Supplementary Figs. 27, 28). LMNP size ranges are judiciously selected for DEAs to avoid rupture of oxide layers, allowing PULM nanocomposites to maintain its insulative properties under high strains[51,52]. With increasing LMNPs (15, 33, and 50 wt%), circular buckling DEAs showed ~30, 25, and 20% reduction to its driving electric fields to achieve ~30% area strain (Fig. 4b). Drastic reduction to the breakdown strength was encountered at higher concentrations (33 and 50 wt%). Correlating with our findings and Pan et al.[27], the drop in breakdown field strength at high fillers concentration can be attributed to the disruption of the soft segment crystallinity and carboxyl hydrogen bonds in the elastomer that leads to added electrons

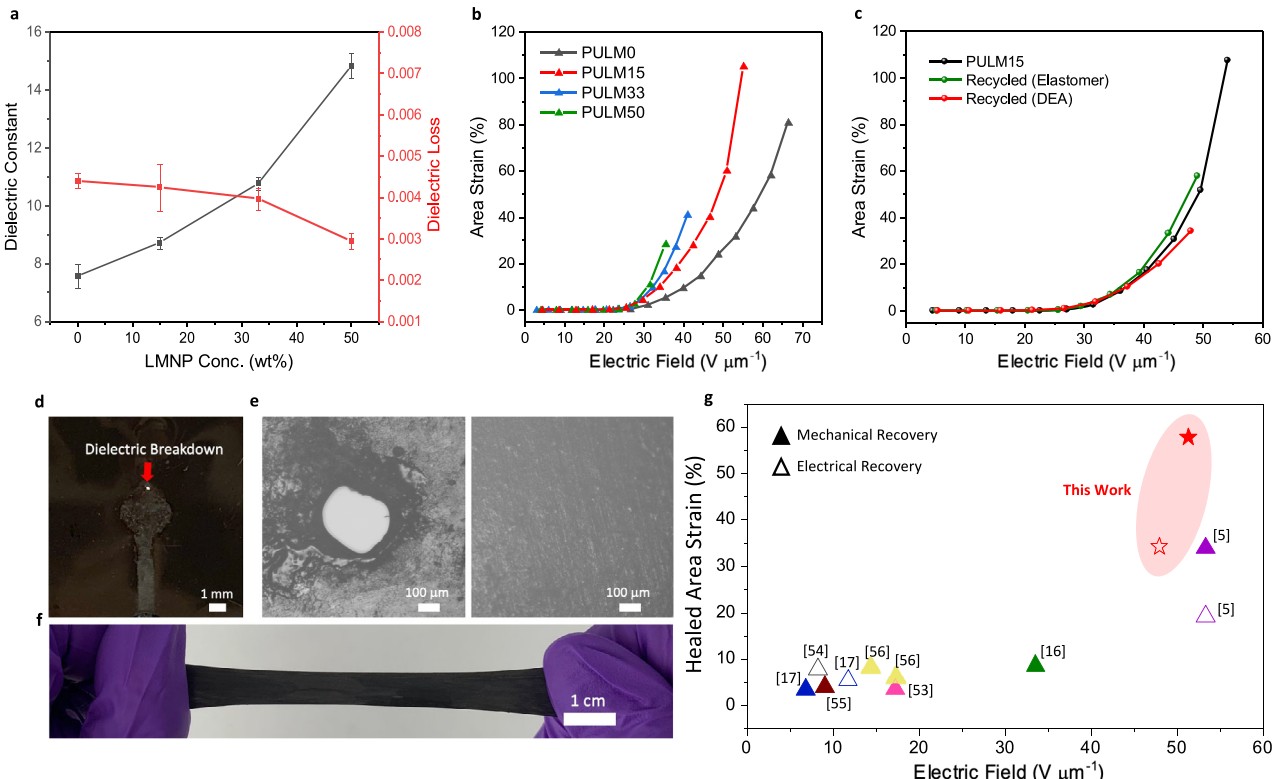

**Fig. 4 | Performance of resilient DEAs. a** Dielectric constant and loss of PULM nanocomposites. Error bars are the standard deviation of five independent samples. **b** Representative area strains achieved by PULM nanocomposite DEAs. Area strains calculated are based on geometric relations from out-of-plane displacements of the buckled electroactive regions. **c** Representative area strains achieved by PULM15, recycled PULM15 after mechanical damages (cut into pieces) and electrical damage (dielectric breakdown). **d** Hole observed at the electrode area is attributed to dielectric breakdown. **e** Optical microscope image of dielectric breakdown site and same site after undergoing the recycling process. **f** Recycled film after dielectric breakdown can be pulled up to 250% without indicating any holes and damages. **g** Ashby chart summarizing the healed area strains vs electric fields of various DEA works that can recover its performance after mechanical (solid symbol) or electrical damage (hollow symbol) through self-healing or recycling processes. Actuated radial strains have been approximated to area strains.

pathways. Thus, PULM15 gives the optimal ratio of LMNPs, attaining a maximum area strain of 105% at 55 V μm⁻¹.

In addition to achieving the toughest DEA (Supplementary Table 2), the dynamic hydrogen bonds present within PULM15 allow a recyclable DEA to be realized. The performance of the DEA was evaluated by recycling PULM15 after mechanical damage (cutting into pieces) and electrical damage. After hot-pressing the cut pieces of PULM15, the recovered dielectric layer, achieved an area strain of 57.1% (Fig. 4c). Subsequently, DEAs were recovered through similar procedures after undergoing dielectric breakdown, evident from the small hole formed from localized heating at large electric fields (Fig. 4d). After breakdown occurred, electrodes were removed and the breakdown region was pinched together before hot pressing, resulting in complete coverage of damages as shown from optical microscopic images (Fig. 4e). No damages were observed after stretching the recovered dielectric elastomer by ~250% strain (Fig. 4f) at which the recovered DEA after electrical damage achieved area strains of 34.2%. Compared to the state-of-art self-healing DEAs (Fig. 4g)[5,16,17,53–56], the DEAs in this work could achieve larger recovered area strains. The higher driving electric fields required can be attributed to the increased elastic modulus which is addressed through co-stimulation of NIR light and electric field.

## Photothermal modulated DEAs

Photothermal effects of LMNPs were subsequently used to augment actuation through photothermal softening that lowered the modulus[48,57]. Owing to the high transparency of silver nanowire electrodes to NIR light (Supplementary Fig. 29), PULM dielectric

elastomers could be softened easily through the direct illumination of the DEA. By modulating the NIR light intensity between 0.2 and 0.6 W cm⁻² while simultaneously applying the electric field, the degree of modulus reduction can be controlled on-demand, shifting area strains towards lower driving electric fields (Fig. 5a, b and Supplementary Movie 1). However, at higher NIR intensities, the breakdown strength gradually reduces due to the ease of electromechanical instabilities that dominate at lower moduli[58]. In contrast, PULM0 is not affected by NIR illumination due to its negligible photothermal characteristics (Supplementary Fig. 30). Under simultaneous NIR light and electric field (~30 V μm⁻¹) cycling at 0.1 Hz, actuation drifts could be suppressed below 5% area strains at intensities of 0.2 and 0.4 W cm⁻². As NIR light intensified further to 0.6 W cm⁻², actuation drifts drastically increased due to considerable viscoelastic contributions at higher temperatures, indicating the limiting factor of using high intensities for dynamic actuation (Fig. 5c). Comparing amongst the DEAs with fillers[35,59–66], the favorable combination of insulative, dielectric and photothermal properties of LMNPs in this work has effectively utilized NIR light to augment actuation strains under high breakdown fields (Fig. 5d).

To achieve an actuator with multiple stimuli response to NIR light and electric fields, a bilayer configuration was adopted. A dielectric minimum energy structure (DEMES) was designed to achieve higher bending-mode actuation by adhering a lightly pre-strained dielectric elastomer onto a stiffer passive polyimide layer. Starting with a curved state, the application of a voltage generates a Maxwell pressure that changes the bending angle to a lower curvature to realize a new minimum energy state (Supplementary Fig. 31)[5,67]. Similar to circular

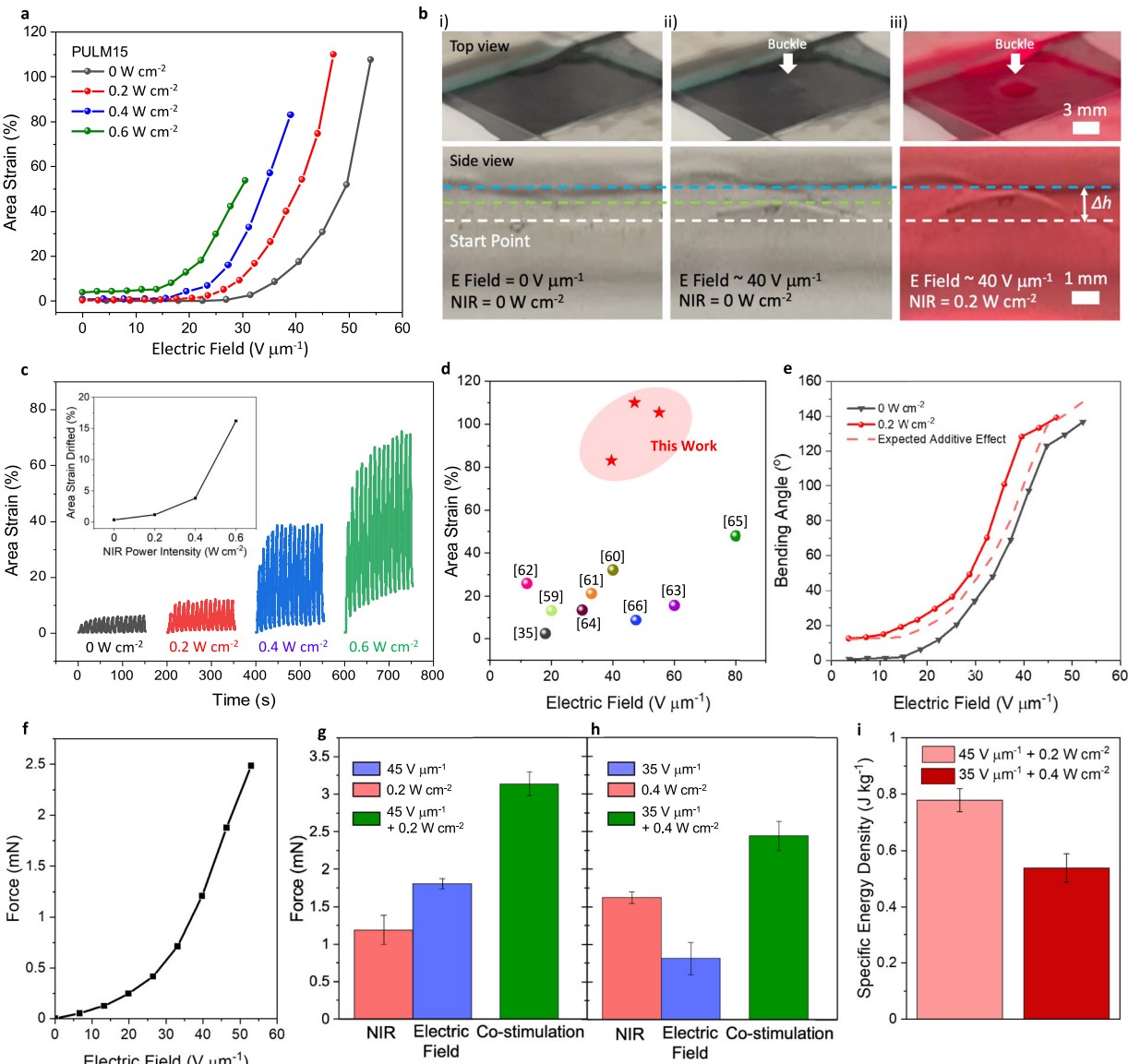

**Fig. 5 | Co-stimulation of PULM15 DEA. a** Representative area strains of PULM15 DEAs driven by electric field ($E$ field) and NIR light. **b** Digital images of top and side views of PULM15 DEA in the (i) original state (ii) solely driven by electric field, and (iii) simultaneously driven by electric field and NIR light. Dotted lines indicate the height of out-of-plane displacements during actuation. $\Delta h$ represents the change in height between the original and co-stimulated state. **c** Cyclic actuation of PULM15 DEA being simultaneously driven by electric field and NIR light at 0.1 Hz. Inset shows the drifted area strains after 15 cycles as a function of NIR light intensity. **d** Ashby chart summarizing the area strains vs electric fields of DEAs that utilize fillers. **e** Representative bending angles of PULM15 DEMES individually driven by electric field and simultaneously driven by electric fields and NIR light intensity of 0.2 W cm$^{-2}$. Dotted line represents expected additive effects based on the addition of bending angles when driven by electric field and NIR light individually. **f** Representative blocking force of PULM15 DEMES **g** Blocking force output of PULM15 DEMES when activated solely by an electric field of 45 V μm$^{-1}$ (blue) or NIR light intensity of 0.2 W cm$^{-2}$ (red), and when co-stimulated by both electric field and NIR light (green). **h.** Blocking force output of PULM15 DEMES when activated solely by an electric field of 35 V μm$^{-1}$ (blue) or NIR light intensity of 0.4 W cm$^{-2}$ (red), and when co-stimulated by both electric field and NIR light (green). **i** Specific energy density of PULM15 DEMES under co-stimulation at various electric fields and NIR light power intensities. All error bars are the standard deviation of three independent samples.

buckling DEAs, PULM15 achieved larger bending angles at a lower electric field due to its improved dielectric constant (Supplementary Fig. 32). In addition, the photothermal effects of LMNPs enabled the DEMES to actuate in response to NIR light due to asymmetric thermal expansion between bilayers (Supplementary Fig. 33). Realized in combination, synergistic effects between both stimuli were evident as bending angles were found to be larger at lower electric fields in comparison to the expected additive effect of individual DEMES and photothermal actuation (Fig. 5e and Supplementary Fig. 34). As investigated earlier, this enhancement can be ascribed to photothermal softening effects that enhanced actuation strains achieved by

DEAs. In addition, blocking forces of PULM15 DEMES under various stimuli conditions were investigated. At a driving electric field of 53 V μm$^{-1}$, a maximum blocking force of 2.5 mN (0.25 gf) was achieved (Fig. 5f), which is twice the average weight of the actuator (0.125 g). Higher blocking forces was further obtained at 3.1 mN under co-stimulation of NIR light and electric field at 0.2 W cm$^{-2}$ and 45 V μm$^{-1}$, respectively (Fig. 5g). On the other hand, the driving electric field to achieve 2.5 mN was lowered to 35 V μm$^{-1}$ when co-stimulated with 0.4 W cm$^{-2}$ (Fig. 5h). Based on the blocking force and bending displacements, a larger maximum energy density of 0.78 J kg$^{-1}$ was achieved during co-stimulation compared to 0.59 J kg$^{-1}$ that when

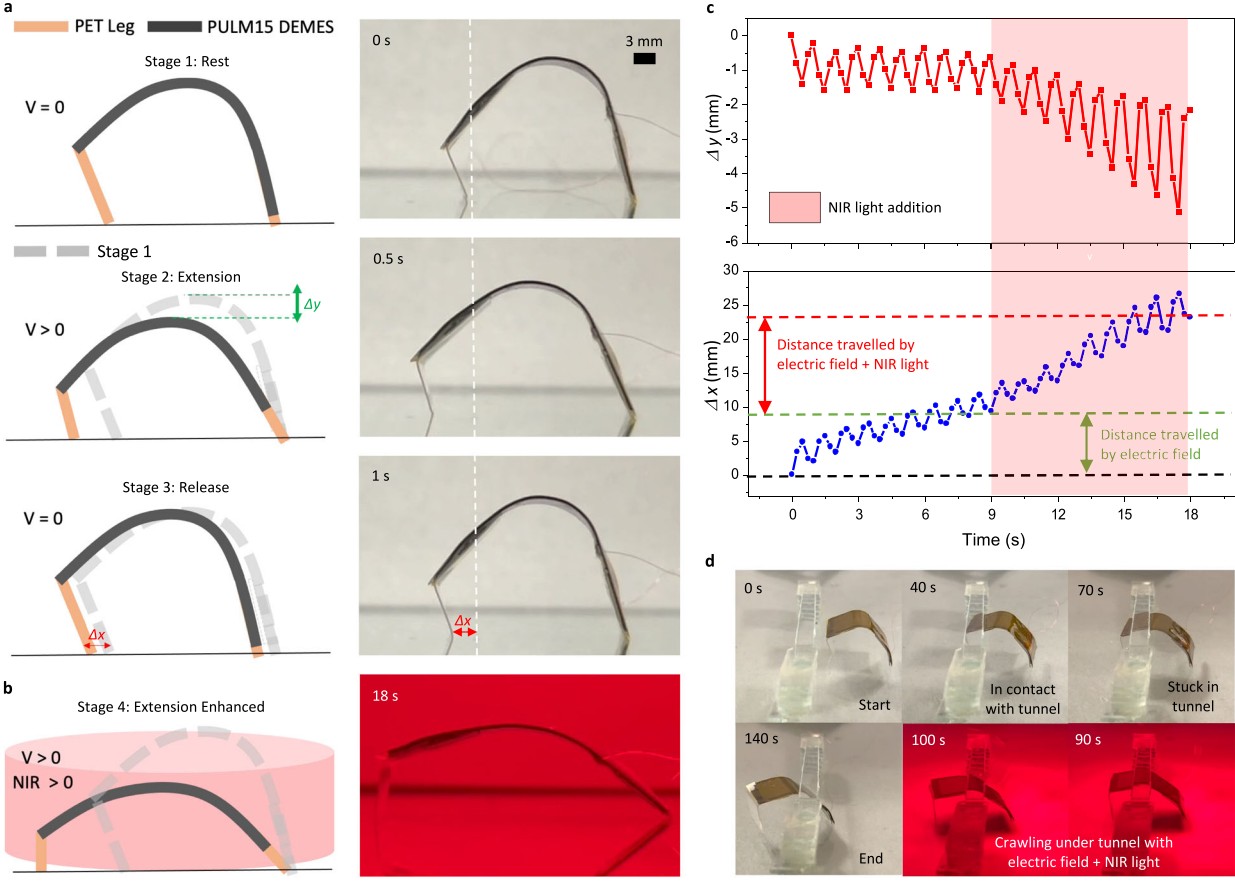

**Fig. 6 | Locomotion analysis of DEMES crawler. a** Schematic and digital images of the working principle of two-anchor crawling locomotion of a DEMES crawler. When a voltage ($V$) is applied, the crawler flattens and extends to give a vertical ($\Delta y$) and lateral displacement ($\Delta x$). After the voltage is removed, asymmetric friction from the bent front PET leg enables the crawler to crawl forward. **b** Schematic and digital image of the usage of NIR light as a secondary control to increase the stride of the crawler, with larger $\Delta x$ and $\Delta y$. **c** $\Delta y$ and $\Delta x$ of two-anchor crawling loco-motion as a function of time. First, the crawler was driven by an electric field of 35 V μm$^{-1}$ at 1 Hz. After which, NIR light (0.4 W cm$^{-2}$) was applied continuously while the crawler was driven by the same electric field. When NIR light was applied, the enhanced stride led to a larger distance traveled. **d** Digital image of the crawler driven by 35 V μm$^{-1}$ at 1 Hz to crawl under a low tunnel (width 50 mm × length 5 mm × height 18 mm). When the crawler comes into contact with the tunnel (40 s), it first attempts to adapt its body to crawl under the tunnel based on its soft and flexible nature (40–70 s). Due to the larger height of the crawler, it eventually gets stuck in the tunnel. When NIR light is applied, the crawler is able to release itself to crawl through the tunnel due to a larger vertical displacement generated during crawling.

solely driven by electric field (Fig. 5i and Supplementary Figs. 35, 36). Compared to other DEA unimorphs (Supplementary Table 3), the energy density of single-layer PULM15 actuators is attributed to its enhanced ability to bend across large distances. These results highlight the versatility of the nanocomposite actuator to respond to multiple stimuli and even achieve synergistic actuation when applied simultaneously.

These enhancements provided through co-stimulation are applied to two-anchor crawling of a DEMES crawler. Stride perfor-mances of the crawler were first being evaluated by locking the rear leg to minimize slippage effects and to ensure that the crawler remained within the measurement range of displacement sensors (Supplemen-tary Fig. 37). Two conditions of co-stimulation were explored. First, NIR light was applied at the same frequency as the electric field. Second, the NIR light was applied continuously while the electric field was driven at a frequency. NIR light applied continuously led to increased flattening and longer strides but was accompanied by larger actuation drifts (Supplementary Fig. 38). Similar effects were observed when NIR light intensities were increased (Supplementary Fig. 39). These find-ings are attributed to the higher generated temperatures at longer exposure times. Upon removal of NIR light, the crawlers could gra-dually recovered their shape while being operated by electric field alone or when stimuli were removed. The speed of recovery is highly

dependent on temperatures generated within the elastomer at which shorter recovery times were observed at lower NIR light intensity or when NIR light was applied at a frequency. This highlight the versatility of co-stimulation through modulating NIR light exposure times to control performances of DEMES crawler.

To achieve locomotion, an electric field is applied to enable the crawler to flatten and extend its front leg forward while anchoring its rear leg. Conversely, after removing electric fields, the crawler retains its original state, at which the front leg pulls the crawler forward to achieve a net displacement (Fig. 6a). Driven only by electric fields of 35 V μm$^{-1}$, 1 Hz signal, vertical displacements reduced (crawler flat-tened) by ~1.1 mm, translating to a horizontal motion or lateral dis-placement ($\Delta x$). By repeating this process over multiple cycles and introducing a bent PET front leg for frictional bias, the crawler could translate forward. NIR light can be adopted as a secondary control stimulus to enable longer strides from synergistic actuation effects (Fig. 6b and Supplementary Movie 2). To investigate the impact of NIR light on two-anchor crawling, lateral ($\Delta x$) and vertical displacements ($\Delta y$) were measured with time (Fig. 6c). A lower lateral displacement ($\Delta x$) over time can be attributed to frictional slips that are often observed in crawling-based locomotion[68]. Upon continuous illumina-tion of NIR light at 0.4 W cm$^{-2}$, vertical displacements ($\Delta y$) increased further with each cycle due to increased heat generation with time. As

a result, longer strides are realized to travel further under a fixed period. Enhanced flattening of the crawler under co-stimuli of NIR light and electric field offers shape adaptation capabilities at narrow gaps (Fig. 6d and Supplementary Movie 3). As the crawler (height 22 mm) first encounters a low tunnel (height 18 mm), the soft and flexible nature enables the crawler to adapt to the narrow space, continuing to move forward. However, as the crawler continues to squeeze through the tunnel, contact between the crawler and tunnel ceiling is increased. As a result, larger generated frictional force causes the crawler to be eventually stuck midway in the tunnel. Consequently, NIR light was applied to enable increased flattening and minimized contact with the tunnel ceiling, allowing the crawler to be released and advance forward. In comparison to the use of increasing voltage input to increase the flattening of DEA crawlers[69], our approach may reduce the probability of dielectric breakdown or enable the design of future untethered soft robots with commercial portable power supplies[48].

## Discussion

The introduction of LMNPs to carboxyl polyurethane has enabled high performing resilient DEAs. Carboxyl functionalities impart a strong and dynamic physical crosslinked network to the elastomer while LMNPs provides high dielectric constants and photothermal properties. These nanocomposites display high mechanical toughness, self-healing capabilities and recyclability. Through thermal enhanced self-healing with NIR light, localized healing can take place, avoiding the need to heat the entire device that may contain thermal-sensitive components. Furthermore, due to the disruptive role of LMNPs to the hydrogen bonding between polymer chains, lower energy requirements to recover their properties through recycling are realized. Broken pieces of elastomers could be recycled and re-applied as a DEA, achieving area strains up to 57.1% that exceeds state-of-art self-healing DEAs. As mechanical enhancements often comes with increased stiffness that is detrimental to actuation, the concept of co-stimulation is introduced to achieve high performing and resilient DEAs. Photo-thermal heating lowers the elastic modulus of the elastomer for higher actuation strains to be achieved under electric fields. Unlike the addition of plasticizer and molecular designs, photothermal heating provides a reversible approach to recover its favorable mechanical properties after cooling to maintain damage resistance. Compared to other metal and carbon-based photothermal fillers, the insulative gallium oxide shell that is distinctive of liquid metal prevents premature dielectric breakdown, unlocking the potential of electric field and NIR light co-stimulation. Furthermore, unlike other dielectric elastomer composites, the PULM nanocomposites remain highly stretchable even at higher loadings. With bilayer structures, the nanocomposites can be utilized in a photothermal actuator and a DEA individually, and in combination display a synergistic actuation due to softening effects. The enhancements with NIR light are further demonstrated through a crawler, where activation of NIR light lengthened its stride and increased its shape adaptability to crawl through narrow gaps. Thus, this work has the potential to expand the application of DEAs within soft robotics, providing on-demand modulation without complex electrode patterns or increasing driving voltages. Future works will further explore applying localized illumination of NIR light for greater shape morphing and improve heat dissipation effects. Therefore, through our material design, we achieve a new generation of resilient soft robots with long operational lifetimes and high actuation performance.

## Methods

### Synthesis of carboxyl polyurethane

Carboxyl polyurethane was synthesized in a one-step polymerization process according to the procedure shown in Supplementary Fig. 1. Before polymerization, PTMG (8.3 mmol) was vacuum dried for 1 h at 110 °C under Ar atmosphere to remove moisture residues. Subsequently, DMF (24 ml), IPDI (20.7 mmol), DMPA (10.7 mmol) and DBTDL (10 μl) were added into the vessel and stirred for 24 h at 80 °C under argon atmosphere. After the reaction, methanol was added and stirred for 30 min to quench the remaining isocyanate groups. White precipitate liquid that formed settled to the bottom, at which the clear upper region was decanted. The remaining amount of methanol was removed through vacuum drying and the resultant solution was re-dissolved in DMF.

### Synthesis of PULM nanocomposites

LM nanoparticles were prepared through ultrasonication of the bulk LM alloy (GaInSn) in DMF (0.15 g/ml) at 300 W for 1 h. The resultant LM solution was then mixed with carboxyl polyurethane at different weight ratios. Finally, PULM nanocomposite films were obtained by pouring the mixture into a petri dish, followed by drying at 80 °C for 12 h. Sample names were coded as PULMX at which X refers to the weight ratio between LM and carboxyl polyurethane.

## Data availability

All data needed to evaluate the conclusions in the paper are present in the paper and/ or the Supplementary Materials. Additional data related to this paper may be requested from the corresponding author.

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

## Acknowledgements

M.W.M.T. acknowledges the scholarship awarded by the Nanyang Technological University, Singapore. We acknowledge the financial support from the Singapore Ministry of Education, AcRF Tier 1 Grant RG63/20.

## Author contributions

M.W.M.T. and P.S.L. designed the research paper. P.S.L. supervised the overall project. M.W.M.T. synthesized the material. M.W.M.T., B.H., G.T., and X.G. characterized the materials and analyzed the results. M.W.M.T. designed the actuator and measured its performance. M.W.M.T. and P.S.L. wrote the manuscript and supporting information. All authors edited and agreed with the manuscript.

## Competing interests

The authors declare no competing interests
