## [Peer Review File · Nature Communications]

Photothermal modulated dielectric elastomer actuator for resilient soft robotsReviewers' Comments:

Reviewer #1:

Remarks to the Author:

Tan et al present a materials architecture of liquid metal nanoparticles embedded carboxyl polyurethane elastomer for dielectric elastomer actuators with multi-responsiveness to electric fields and near infrared light. The inclusion of liquid metal nanoparticles enhances the NIR absorption and thermal conductivity of this materials system, which enables the responsiveness to NIR light and better recyclability. Though, the authors have done systematic characterization and provided abundant experimental results, I still don't recommend to published on Nature Communications with following reasons.

General comments:

1. The authors present this work as dielectric elastomer actuator, however, the most of results characterized the new dielectric elastomer system instead of actuators. The manuscript may attract audiences in material engineering community instead of the targeted audiences in actuation and soft robotics community.
2. The presented results should be more sharply for the main subject (actuator). For example, most dielectric elastomer related characterizations were conducted on circular configuration of actuator and presented area strain (e.g. Figure 3 b,c,e), showing barely help to the actual actuators and demos with bilayer configuration. The authors should discuss more on actuator design and characterization if the theme is DEA, and set the logical connection between the experimental results of dielectric materials system and the design and performance of dielectric elastomer actuators.
3. The intro needs careful revision for better description of novelty and precise comments on the research background.

Other comments and concerns:

1. In Mechanical and photomechanical properties section, the authors did mechanical test up to very high strain (>2000%). However, the DEAs characterized in this work worked in medium strain range (<100% area strain). I will recommend the authors providing the mechanicals system property in medium strain range especially for cyclic and recovery test.
2. Figure 3d is hard to follow. It's better to retake the images.
3. How did NIR apply on the functional material (dielectric elastomer) of DEA through the stretchable electrodes? Is because the sliver NWs electrodes transparent to NIR?
4. I will suggest adding schematics to show the design of DEAs.

Reviewer #2:

Remarks to the Author:

In this manuscript, the authors show a nanocomposite based dielectric elastomer actuator (DEA) with responsiveness to electric fields and NIR light. This permits the association of the mechanical properties with NIR radiation. The nanocomposite DEA comprises a polymer-based elastomer with liquid metal NPs as a filler. In a soft crawler scenario, the co-stimulation (electrical/optical) leads to the light-mediated modulation of its mechanical properties, and the soft crawler changes its locomotion behavior. On top of this, the DEA can be healed by the very same (optical) stimulus, while the nanocomposite actuator can be (partially) recycled by a hot press process at relatively low thermal budgets (~100 oC for 15 mins). The results represent an interesting collection of materials properties for soft actuators. The manuscript is also well organized. I believe that the manuscript could be suitable for publication in the journal. Prior to this, I have a few comments:

1. It seems that there are 2 different concepts in the manuscript. One is co-stimulation and the other one recyclability. The motivation and benefits of combining recyclability in with soft robotics/actuators in not obvious. In the manuscript it sounds like a combination of properties with not so well-constructed arguments.
2. Why there is saturation of photothermal conversion as a function of the LMNPs concentration?

3. The NIR light intensity for the photothermal softening of the DEA is between 0.2 to 0.8 W cm⁻². How this intensity correlates with real physical values? Is it too high for actuation? Please comment on that. What would be the strategy to reduce the needed NIR light intensity?
4. What is the nature of the nonlinearity of the synergetic phenomena (photothermal and mechanical)?
5. In the soft crawler demonstrator, there is a certain time for heat dissipation and the recovery of the non-synergetic function (also shown at Fig. 2). This recovery time could potentially introduce hysteretic locomotion behaviors. What is the strategy to alter (reduce) this recovery time?
6. The authors claim that the nanocomposite soft actuator exhibits self-healing abilities. This is not self-healing, as it is enabled by external cues. This is external stimuli-mediated healing. Please rephrase wherever necessary.

Reviewer #3:

Remarks to the Author:

This is rather an interesting article by Pooi See Lee et al. on a dual-mode soft actuator that shows promising features such as recyclability and self-healing. The novelty of the work is the material design and integration of both electrical field and photothermal stimulations. While the embedded liquid metal (LM) nanodroplets play a key role in tuning the functional properties of the composites, the carboxyl polyurethane matrix offers uniqueness and distinguishes this work. However, there are some similarities between this work and previous studies as it is discussed below. Overall, the paper is well-written and well-presented, and most of the claims are supported by experimental results and supplementary information. The unique material design and potential application in soft robotics will attract a broad range of audience and I believe it can be a good fit for the Nature Communications journal. Here are my specific comments that I hope will improve the paper:

1. As it was mentioned earlier, the working mechanism of the soft actuator and integration of LM nanoparticles have been explored previously. For instance, a recent study by Xuechang Zhou et al. shows the use of LM-liquid crystal composites with recyclable, weldable, and actuation behaviors. They also used NIR and electric field for actuation, as well as making a crawling robot and demonstrating recyclability, and so on. Read more: Lv, P., Yang, X., Bisoyi, H. K., Zeng, H., Zhang, X., Chen, Y., ... & Li, Q. (2021). Stimulus-driven liquid metal and liquid crystal network actuators for programmable soft robotics. *Materials Horizons*, 8(9), 2475-2484.
2. The authors have done a great job for the motivation of this work and presenting the results, however the literature review on the materials aspect is weak. If the novelty of this study is introducing a novel LM-polymer nanocomposite with enabling properties, then a brief background needs to be given to the readers. There are recent review articles on liquid metal nanocomposites, multifunctional liquid metal elastomer composites, and liquid metal materials for soft robotics that are highly related to the study in terms and can support certain claims in the manuscript.
3. Page 4, line 164-166: "Elastic modulus and mechanical toughness increased with LMNPs addition at 15 and 33 wt%, due to the effective load transfer between the elastomer matrix and the presence of gallium oxide shells which are stiffer (Fig. 2b) 35,36". Can you clarify the meaning of effective load transfer and its relationship with stiffness? This increase is because of the solid gallium oxide shell and the effective elastic modulus can be even predicted using a double inclusion micromechanics model for liquid metal nanocomposites. Reference 35 on graphene (a solid filler) – TPU composites appears out of place unless the interface failure is being studied.
4. In Figure 3e, it looks like increasing the NIR light intensity to 0.6 W/cm² increases the actuation significantly, but it does not fully recover like the case for 0.2 W/cm². Wouldn't this cause an issue in recovering the original shape? This irreversible behavior is also observable for the crawling robot demo in Figure 4c. The Δy keeps increasing but it does not fully recover like the actuation in the

first half without NIR light.

5. Please provide details on how the bending angles are calculated and comment on how robust your approach is. Based on the schematic in the supplementary figure 22, the bending angle is highly dependent on the selection of the two lines and can lead to a significant variation in the bending angle. I'm also curious to see how this bilayer actuator can achieve a bending angle of 120-140 degrees.

6. In DSC results (Figure 1f and Figure S8), there are additional peaks for LM filled PU. These peaks are around -27 degrees C in the heating cycle as well as peaks at around -50 and -65 degrees C in the cooling cycle. Are these peaks related to the melting/freezing of LMNPs? It could be like the supercooling effect recently reported for liquid metal composites. Please elaborate more on this point in the manuscript.

Minor suggestions:

1. The x-axis in Figure 4c is time (ms). Is the time measured in milliseconds or it's a typo? The speed in Movie S1 indicates it should be in seconds.

2. Dashed lines in Figure 3h says "Theory". This can be misleading unless there is a theoretical model to predict the actuation. You may consider rephrasing this.

3. Page 7, line 287: "Fig. 4d and Supplementary Movie 4" This must be Movie 3 because there were only three SI movies.

Response Letter:

Recyclable and photothermal modulated dielectric elastomer actuator for soft robotics

*Matthew Wei Ming Tan, Hyunwoo Bark, Gurunathan Thangavel, Xuefei Gong, Pooi See Lee**

We would like to thank the editors of *Nature Communication* for handling our manuscript. Our gratitude is also extended to the reviewers for their constructive comments. All the comments raised by the reviewers have now been fully addressed in this revision. The comments from the reviewers are reprinted in blue Times New Roman. Our corresponding responses are marked in black Times New Roman and the changes made to the manuscript are highlighted in yellow.

REVIEWER COMMENTS

Reviewer #1 (Remarks to the Author):

Tan et al present a materials architecture of liquid metal nanoparticles embedded carboxyl polyurethane elastomer for dielectric elastomer actuators with multi-responsiveness to electric fields and near infrared light. The inclusion of liquid metal nanoparticles enhances the NIR absorption and thermal conductivity of this materials system, which enables the responsiveness to NIR light and better recyclability. Though, the authors have done systematic characterization and provided abundant experimental results, I still don't recommend to published on Nature Communications with following reasons.

General comments:

1. The authors present this work as dielectric elastomer actuator, however, the most of results characterized the new dielectric elastomer system instead of actuators. The manuscript may attract audiences in material engineering community instead of the targeted audiences in actuation and soft robotics community.

Thank you for the comments. In addition to the characterization of the dielectric elastomer system, we have expanded the actuator characterizations such as the force and energy density of the actuator, and strides of the soft crawler. With these changes, we have addressed reviewer concerns on the scope of work. Accordingly, the titled has been adjusted to “Photothermal modulated dielectric elastomer actuator for resilient soft robots” that would cater to the interest of the soft robotic community.

2. The presented results should be more sharply for the main subject (actuator). For example, most dielectric elastomer related characterizations were conducted on circular configuration of actuator and presented area strain (e.g. Figure 3 b,c,e), showing barely help to the actual actuators and demos with bilayer configuration. The authors should discuss more on actuator design and characterization if the theme is DEA, and set the logical connection between the experimental results of dielectric materials system and the design and performance of dielectric elastomer actuators.

To further evaluate the performance of bilayer configurations of DEAs, additional experiments to determine their blocking force outputs were conducted. The blocking force output of PULM15 bilayer actuator with increasing electric field was measured (Fig. 5f). Maximum blocking force of the PULM15 was found to be 2.5 mN (0.25 gf) at $53 \text{ V } \mu\text{m}^{-1}$, that is twice the average weight of the actuator (0.125 g). When driven by NIR light at 0.2 and 0.4 W cm^{-2} , a blocking force of 1.1 and 1.6 mN was achieved. Under co-stimulation of NIR light and electric field at 0.2 W cm^{-2} and $45 \text{ V } \mu\text{m}^{-1}$ respectively, a larger blocking force of 3.1 mN was achieved (Fig. 5g). Based on the calculation utilizing the force and bending displacements of the actuator tip, a maximum energy density of 0.59 J kg^{-1} was calculated for PULM15 bilayer DEAs when driven by electric field (Fig. 5i). This is approximately three times larger than the lower bound of biological muscles (0.2 J kg^{-1})¹. However, to reach the upper bound of biological muscles (40 J kg^{-1}), stacking approaches can be adopted in the future². Through co-stimulation with NIR light and electric field, the energy density can be enhanced further enhanced to 0.78 J kg^{-1} , highlighting the potential of this strategy for actuation.

Additional discussion and schematics have been provided on the design of the various actuator configurations in supplementary Fig 1-2 and supplementary note 1.

On Page 8 of Main manuscript:

In addition, blocking forces of PULM15 DEMES under various stimuli conditions were investigated. At a driving electric field of $53 \text{ V } \mu\text{m}^{-1}$, a maximum blocking force of 2.5 mN (0.25 gf) was achieved (Fig. 5f), that is twice the average weight of the actuator (0.125 g). Higher blocking forces was further obtained at 3.1 mN under co-stimulation of NIR light and electric field at 0.2 W cm^{-2} and $45 \text{ V } \mu\text{m}^{-1}$ respectively (Fig. 5g). On the other hand, the driving electric field to achieve 2.5 mN was lowered to $35 \text{ V } \mu\text{m}^{-1}$ when co-stimulated with 0.4 W cm^{-2} (Fig 5h). Based on the blocking force and bending displacements, a larger maximum energy density of 0.78 J kg^{-1} was achieved during co-stimulation compared to 0.59 J kg^{-1} that when solely driven by electric field (Fig. 5i and Supplementary Fig. 35-36). Compared to other DEA unimorphs (Supplementary Table 3), the energy density of single layer PULM15 actuators is attributed to its enhanced ability to bend across large distances.

Fig. 5f Representative blocking force of PULM15 DEMES **g** Blocking force output of PULM15 DEMES when activated solely by an electric field (E Field) of $45 \text{ V } \mu\text{m}^{-1}$ (blue) or NIR light intensity of 0.2 W cm^{-2} (red), and when co-stimulated by both electric field and NIR light (green). **h**. Blocking force output of PULM15 DEMES when activated solely by an electric field of $35 \text{ V } \mu\text{m}^{-1}$ (blue) or NIR light intensity of 0.4 W cm^{-2} (red), and when co-stimulated by both electric field and NIR light (green). **i** Specific energy density of PULM15 DEMES under co-stimulation at various electric fields and NIR light power intensities.

On Page 3-4 of Supporting Information:

The blocking forces were measured using a load cell (LSB200, Futek) in contact with the free end of the DEMES. Specific energy densities (E_m) were calculated using equation S3. F_b represents the blocking force, δ represents the actuator bending displacement and m is the mass of the actuator. δ can be derived from the bending angles (θ) based on equation S4. L represents the active length of the DEMES.

$$E_m = \frac{(F_b \delta)}{2m} \quad (\text{S3})$$

$$\delta = \theta \left(\frac{\pi}{180} \right) L \quad (\text{S4})$$

On Page 28 of Supporting Information:

Supplementary Fig. 35 Bending displacement of PULM15 DEMES at various electric field

Supplementary Fig. 36 Specific energy density of PULM15 DEMES driven solely by **a** electric field and **b** NIR light.

Supplementary Table 3 Comparison between actuation of unimorph single layer DEAs

Material	Electric Field (V μm ⁻¹)	Bending Angle (°)	Bending Displacement (mm)	Blocking Force (mN)	Specific Energy Density (J kg ⁻¹)	Ref
PULM15	53	136	60.4	2.5	0.59	This Work
PULM15 (Co-stimulation)	45	139	66.0	3.1	0.78	This Work

0.2 W cm ⁻²)						
VHB4905 ^a	10	NA	0.5	2	NA	30
BaTiO ₃ /silicone sealant and elastomer	11	NA	9.8	17.3	0.13	31
Silicone Nusil CF19-2186	63	~29.3	NA	~1.19	NA	32
D0	3	40	14.0	20.2	2.68	33
D20	3	22	7.7	9.7	0.75	33
Liquid metal-silicone elastomer	53	NA	NA	~21.6	NA	34

NA: Not available; ~ estimated from figures

^a 8 V μm⁻¹ electric field applied for bending displacement, 10 V μm⁻¹ for blocking force

3. The intro needs careful revision for better description of novelty and precise comments on the research background.

Thank you for your feedback, the introduction has been revised to better justify the background and novelty behind this work. In addition, the contents of the manuscript have been re-organized to provide better coherence of the work.

In this work, co-stimulation is applied to enable soft robots to achieve both resilience and high actuation performance. By enabling the soft actuators to be resilient under harsh and unpredictable environment through damage resistance and recovery, their operational lifetime can be significantly prolonged. Damage resistance can be achieved through utilizing strong and tough elastomers that require larger forces to break. On the other hand, damage recovery can be achieved through self-healing and recyclability. However, the mechanical enhancements used to improve the resilience of soft robots often lead to an increased stiffness that compromises the actuation capabilities. Therefore, our work introduces co-stimulation to DEAs to achieve resilient soft robots with high actuation performances.

On Page 2-3 of Main manuscript:

Soft robots are preferred... When damage occurs, materials imparted with self-healing and recyclability enable soft devices to be easily recovered for continued functionality^{4,7}. This recovery process should be of low-energy to be sustainable and economically favorable for greater adoption⁸.

While previous works have made DEAs healable for prolonged lifetime^{16,17}, imparting high strength and toughness to these actuators remains to be achieved. ... To achieve this, this work introduces the concept of co-stimulation.

Other comments and concerns:

1. In Mechanical and photomechanical properties section, the authors did mechanical test up to very high strain (>2000%). However, the DEAs characterized in this work worked in medium strain range (<100% area strain). I will recommend the authors providing the mechanicals system property in medium strain range especially for cyclic and recovery test.

Thank you for the recommendation. To better correlate the actuation and the mechanical system properties, additional cyclic and recovery tests were performed at limiting strains of 100% and 500%. By selecting these strain regimes, greater insights on the mechanical system properties are provided at very high (1000%), high (500%) and medium (100%) strain regimes.

Based on Fig. 2c, larger hysteresis are observed at increasing strain limits for all PULM films owing to the rupture of strong carboxyl hydrogen bonds at high strains that dissipate energy. It is established that PULM33 was fully recovered at medium strain regimes (Fig. 2d). This findings can be attributed to lower amount of broken hydrogen bonds at medium strain ranges that allow the complete recovery of PULM33.

On Page 5 of Main manuscript:

With higher interfacial friction between LMNPs and polymer chains, hysteresis losses increased with LMNP content at strain limits of 100, 500 and 1000% (Fig. 2c). At increasing strain limits, energy dissipation effects are more pronounced as stronger carboxyl hydrogen bonds are being ruptured⁵. To determine the recoverability of the elastomers, elastomers were left for a rest period of 180 mins after cycling at similar strain limits (Fig. 2d and Supplementary Fig. 15). At medium strains of 100%, PULM0, PULM15 and PULM33 maintained high recovery ratios while PULM50 was unable to recover its performance due to the disruption imposed by LMNPs against the reformation of broken hydrogen bonds between polymer chains.

Fig 2. **c** Hysteresis area when PULM nanocomposites are pulled to 100, 500 and 1000% strain after the first cycle of cyclic stress strain measurements. Inset shows the hysteresis area at 100% strain. **d** Recovered hysteresis area of PULM nanocomposite cyclic stress strain curves after a 1 min and 180 mins delay at 100% strain.

On Page 16 of Supporting Information :

Supplementary Fig. 15 Recovered hysteresis area of PULM nanocomposite cyclic stress strain curves after a 1 min and 180 mins delay at strain limits of **a** 500% and **b** 1000%.

2. Figure 3d is hard to follow. It's better to retake the images.

To better depict the buckling actuation, top view and side view of the actuation has been included for Fig. 3d (now Fig 5b).

Fig 5 b. Digital images of top and side views of PULM15 DEA in the (i) original state (ii) solely driven by electric field, and (iii) simultaneously driven by electric field and NIR light. Dotted lines indicate the height of out-of-plane displacements during actuation

3. How did NIR apply on the functional material (dielectric elastomer) of DEA through the stretchable electrodes? Is because the silver NWs electrodes transparent to NIR?

Thank you for the reviewer's feedback. NIR light was directly illuminated onto the entire DEA composed of the dielectric elastomer and the electrodes. Yes, the functional material could be activated due to the high transparency of silver nanowire electrodes to NIR light. To support this, UV-Vis measurement was performed on silver nanowires (Supplementary Fig. 29), at which a high transparency of was found at both visible and NIR light wavelengths.

On Page 7 of Main manuscript:

Owing to the high transparency of silver nanowire electrodes to NIR light (Supplementary Fig. 29), PULM dielectric elastomers could be softened easily through the direct illumination of the DEA.

On Page 2 of Supporting Information

Transmittance measurement of silver nanowires spray coated on a glass slide was performed using UV-vis-NIR Lambda 950 with reference to a bare glass slide.

On Page 25 of Supporting Information

Supplementary Fig. 29 UV-vis spectra of silver nanowires from 380 to 2500 nm.

4. I will suggest adding schematics to show the design of DEAs.

Thank you for the reviewer's comment. To provide clarity in the working principle and configuration of the DEA, a schematic (Supplementary Fig. 1-2) and discussion (Supplementary Note 1) is provided.

On Page 5-6 of Supporting Information

Supplementary Fig. 1. Schematic of NIR light enhanced circular buckling DEAs at different states. **a** voltage off and NIR light off, **b** voltage on and NIR light off and, **c** voltage on and NIR light off.

Supplementary Fig. 2. Schematic of fabrication and mechanism of DEMES **a.** Fabrication of DEMES by i) pre-straining the dielectric elastomer, ii) applying silver nanowire electrodes through spray coating, iii) adhering the pre-strain dielectric elastomer to a passive flexible layer and iv) releasing pre-strains that result in a bent state. **b.** Actuation of the DEMES is achieved by applying i) solely electric field, ii) solely NIR light and iii) simultaneously NIR light and electric field for co-stimulation to achieve enhanced bending.

Supplementary Note 1.

Supplementary Fig. 1 shows the working mechanism of pre-strain free circular actuators that is often used to determine the basic performance of DEAs. Pre-strains are avoided as the rigid frames used to maintain the strain minimizes the effects of having a soft body and increases its mass¹. Owing to stress relaxation and fatigue effects from pre-strains, the lifetime of the device is minimized¹, preventing the realization of resilient soft robots. The active region of the pre-strain free DEA is composed of overlapping electrodes that sandwiches the dielectric elastomer. When an electric field is applied, a Maxwell pressure is generated to compress the active region. This results in an expansion of the active region of the DEA against the passive region with no overlapping electrodes. Generation of compressive stresses occurs surrounding the active region that upon exceeding a buckling limit, causes an out-of-plane displacement

(Supplementary Fig 1b)^{2,3}. During the illumination of NIR light, photothermal softening reduces the modulus to allow a larger degree of actuation strain to occur.

Supplementary Fig. 2 illustrates the fabrication and working principle of DEMES. First, mechanical energy is input into the dielectric elastomer by applying a pre-strain (Supplementary Fig 2ai)^{4,5}. When the dielectric elastomer is adhered to the passive layer and removed from constraints, the structure bends into a curved shape as elastic energy is transferred from the elastomer to the passive layer as bending energy to minimize the energy state (Supplementary Fig 2aiv). When electric field is applied to the DEMES, the generated Maxwell pressure adds additional strain energy to the elastomer and releases the bending energy stored within the passive layer to reach a flatter state with a new minimum energy (Supplementary Fig 2bi)^{4,5}. When NIR light is applied, differential thermal expansion between the two layers leads to bending of the actuator⁶. During co-stimulation, apart from the combined effects of photothermal and dielectric elastomer actuation, synergistic effects are observed. This originates from photothermal heating that reduces the modulus of the elastomer to allow a greater input of strain energy from Maxwell pressures and a larger release of bending energy stored to reach a flatter state.

Reviewer #2 (Remarks to the Author):

In this manuscript, the authors show a nanocomposite based dielectric elastomer actuator (DEA) with responsiveness to electric fields and NIR light. This permits the association of the mechanical properties with NIR radiation. The nanocomposite DEA comprises a polymer-based elastomer with liquid metal NPs as a filler. In a soft crawler scenario, the co-stimulation (electrical/optical) leads to the light-mediated modulation of its mechanical properties, and the soft crawler changes its locomotion behavior. On top of this, the DEA can be healed by the very same (optical) stimulus, while the nanocomposite actuator can be (partially) recycled by a hot press process at relatively low thermal budgets (~100 °C for 15 mins). The results represent an interesting collection of materials properties for soft actuators. The manuscript is also well organized. I believe that the manuscript could be suitable for publication in the journal. Prior to this, I have a few comments:

1. It seems that there are 2 different concepts in the manuscript. One is co-stimulation and the other one recyclability. The motivation and benefits of combining recyclability in with soft robotics/actuators is not obvious. In the manuscript it sounds like a combination of properties with not so well-constructed arguments.

Thank you for the feedback. The introduction has been revised to better justify the motivation and novelty behind this work. In addition, the contents of the manuscript have been re-organized to provide better coherence of the work.

In this work, co-stimulation is applied to enable soft robots to achieve both resilience and high actuation performance. By enabling the soft actuators to be resilient under harsh and unpredictable environment through damage resistance and recovery, their operational lifetime can be significantly prolonged. Damage resistance can be achieved through utilizing strong and tough elastomers that require larger forces to break. On the other hand, damage recovery can be achieved through self-healing and recyclability. However, the mechanical enhancements used to improve the resilience of soft robots often lead to an increased stiffness that compromises the actuation capabilities. Therefore, our work introduces co-stimulation to DEAs to achieve resilient soft robots with high actuation performances.

On Page 2-3 of Main manuscript:

Soft robots are preferred... When damage occurs, materials imparted with self-healing and recyclability enable soft devices to be easily recovered for continued functionality^{4,7}. This recovery process should be of low-energy to be sustainable and economically favorable for greater adoption⁸.

While previous works have made DEAs healable for prolonged lifetime^{16,17}, imparting high strength and toughness to these actuators remains to be achieved. ... To achieve this, this work introduces the concept of co-stimulation.

2. Why there is saturation of photothermal conversion as a function of the LMNPs concentration?

Thank you for the question. Mentioned in page 5 of the main manuscript, the photothermal properties of LMNPs is attributed to optically induced resonance. This resonance behaviour occurs from the confinement effect when the wavelength of the exciting light source is larger than size of the nanoparticles^{3,4}. Therefore, the photothermal behaviour is highly dependent on the size and interparticle interaction of the nanoparticles. When the concentration of LMNPs increases, the tendency of aggregate formation is higher. As a result, increasing the concentration beyond 15 wt% does not lead to an increase in the photothermal conversion. The saturation of photothermal conversion can be observed in various works at higher concentrations as well^{3,5}.

On Page 5 of Main manuscript:

The saturation of photothermal conversion can be attributed to the higher tendency of aggregate formation at larger concentrations that minimizes the effective optical resonance behaviour⁴⁵.

3. The NIR light intensity for the photothermal softening of the DEA is between 0.2 to 0.8 W cm⁻² How this intensity correlates with real physical values? Is it too high for actuation? Please comment on that. What would be the strategy to reduce the needed NIR light intensity?

In terms of real physical values, the NIR light intensity can be correlated to the temperature that is generated from the elastomer during illumination. The correlation between the temperature generated, NIR light intensities and exposure time can be observed from Fig. 2f and Supplementary Fig. 22. To determine that temperatures generated are not too high to cause thermal degradation during actuation, thermogravimetric analysis (TGA) measurements are conducted (Supplementary Fig. 20). Based on these measurements, the upper bound temperature can be determined to be at 190 °C at which more than 1% weight change occurs. During actuation, temperatures generated were all below 90 °C at which NIR light irradiation conditions were kept between 0.2 to 0.6 W cm⁻² with a maximum exposure time of 10 s. Thus, these intensities are not too high for actuation.

To reduce the NIR light intensity, the photothermal properties of LMNPs can be enhanced through modifying its surface with gold⁶ or polydopamine⁷. However, these modifications may increase conductivity of the films and will require careful optimization to ensure that dielectric breakdown do not occur easily.

On Page 5 of Main manuscript:

To ensure that photothermal effects did not cause thermal degradation during actuation, upper bound temperatures were determine through thermogravimetric analysis at which less than 1% weight change was observed up to 190 °C (Supplementary Fig. 20).

Supplementary Fig. 20. Thermogravimetric analysis of PULM nanocomposites (0, 15, 33 and 50 wt% of LMNPs) from 50 to 500 °C.

4. What is the nature of the nonlinearity of the synergetic phenomena (photothermal and mechanical)?

The nonlinearity originates from the quadratic effect of electric field on the maxwell pressure that drives electromechanical actuation^{8,9}. This implies that the electromechanical contribution to actuation is greater at larger electric fields. At low electric fields, actuated bending angles are largely contributed to asymmetric thermal expansion between bilayers, leading to small deviations from the expected additive effect. Electromechanical contributions remain low as Maxwell pressures generated are insufficient. At increasing electric fields, the Maxwell pressure generated becomes larger in a quadratic manner to cause greater electromechanical contribution to actuated bending angles. This leads to a non-linear deviation from expected additive effect at increasing electric fields. As the combined contributions that drive actuated bending angles are not constant with increasing electric field, the synergetic phenomena is nonlinear. As the photothermal softening influences the electromechanical contributions to a large extent, a linear shift towards lower driving electric fields is observed with increasing NIR light intensity.

The quadratic effect of Maxwell pressures that drive electromechanical actuation can be observed from the equation below where ϵ_0 is the vacuum permittivity, V is the voltage applied across the dielectric elastomer (DE), z is the thickness of the DE layer and p is the effective Maxwell stress (p) that drives electromechanical actuation.

$$p = \epsilon_0 \epsilon_r \left(\frac{V}{z}\right)^2 = \epsilon_0 \epsilon_r E^2 \quad (R1)$$

5. In the soft crawler demonstrator, there is a certain time for heat dissipation and the recovery of the non-synergetic function (also shown at Fig. 2). This recovery time could potentially introduce hysteretic locomotion behaviors. What is the strategy to alter (reduce) this recovery time?

While the main focus of this work is to highlight the potential capabilities of co-stimulation of electric field and NIR light on DEA performances, as mentioned by the reviewer, a challenge faced is the time taken for heat dissipation that affects the recovery time of the actuators.

To address this, the thermal conductivity of the device can be improved in future works to aid heat dissipation and improve the recovery times. For example, as shown in Figure 2f, when PULM0 and PULM15 were heated to the same temperature, faster cooling times are observed from PULM15. This can be attributed to the higher thermal conductivity of PULM15 (Supplementary Fig. 23) that allows better heat dissipation.

On Page 10 of Main manuscript:

Future works will further explore applying localized illumination of NIR light for greater shape morphing and improve the heat dissipation effects.

6. The authors claim that the nanocomposite soft actuator exhibits self-healing abilities. This is not self-healing, as it is enabled by external cues. This is external stimuli-mediated healing. Please rephrase wherever necessary.

Thank you for the suggestion. Based on Fig. 3a, the nanocomposite is able to recover a mechanical toughness of 1.39 MJ m^{-3} in 30 mins at room temperature, without any external stimulation. The recovered toughness is similar to that of some silicone-based elastomers^{10,11}, indicating that the healed elastomer without external cues has been achieved. Based on this rationale, we would retain the term self-healing.

However, we acknowledge that healing under NIR light should be clarified as thermal enhanced self-healing. We have reorganized Fig. 3a-c to reflect this change.

On Page 6 of Main manuscript:

Owing to the minimal polymer chain mobility, a mechanical toughness of $\sim 1.4 \text{ MJ m}^{-3}$ was recovered at room temperature after 30 mins. To improve the recovery of mechanical properties, NIR light can be used to accelerate thermal diffusion of polymer chains for greater reassociation of bonds across the broken interface. The thermal enhanced self-healing of PULM15...

By increasing the NIR light intensity from 0.2 to 0.6 W cm^{-2} , mechanical toughness could be recovered from 18 to 55 MJ m^{-3} within 30 mins (Fig. 3a). The improvement is attributed to better interpenetration and diffusion of polymer chains across the broken interface at higher temperatures generated. Being a diffusion-dependent process, increasing the healing time to 120 mins allows similar mechanical toughness to be achieved at low NIR light intensity of 0.2 W cm^{-2} (Fig. 3b). Complete healing at the surface of the cut interface can also be observed from optical images (Fig. 3c).

Fig. 3 Self-healing and recyclability of PULM15 nanocomposite **a** Recovered toughness of PULM15 at different NIR light intensity for 30 mins. **b** Recovered toughness of PULM15 at different healing times under NIR light of $0.2\ W\ cm^{-2}$. **c** Optical microscope images of PULM15 damaged (left, 0 min) and healed after 120 min under NIR light of $0.2\ W\ cm^{-2}$.

Reviewer #3 (Remarks to the Author):

This is rather an interesting article by Pooi See Lee et al. on a dual-mode soft actuator that shows promising features such as recyclability and self-healing. The novelty of the work is the material design and integration of both electrical field and photothermal stimulations. While the embedded liquid metal (LM) nanodroplets play a key role in tuning the functional properties of the composites, the carboxyl polyurethane matrix offers uniqueness and distinguishes this work. However, there are some similarities between this work and previous studies as it is discussed below. Overall, the paper is well-written and well-presented, and most of the claims are supported by experimental results and supplementary information. The unique material design and potential application in soft robotics will attract a broad range of audience and I believe it can be a good fit for the Nature Communications journal. Here are my specific comments that I hope will improve the paper:

1. As it was mentioned earlier, the working mechanism of the soft actuator and integration of LM nanoparticles have been explored previously. For instance, a recent study by Xuechang Zhou et al. shows the use of LM-liquid crystal composites with recyclable, weldable, and actuation behaviors. They also used NIR and electric field for actuation, as well as making a crawling robot and demonstrating recyclability, and so on. Read more: Lv, P., Yang, X., Bisoyi, H. K., Zeng, H., Zhang, X., Chen, Y., ... & Li, Q. (2021). Stimulus-driven liquid metal and liquid crystal network actuators for programmable soft robotics. *Materials Horizons*, 8(9), 2475-2484.

Thank you for the reviewer's feedback. The novelty of this work features the usage of liquid metal nanoparticles to achieve 1) The synergistic effect between electric field and NIR light to achieve enhanced actuation of DEAs and 2) the reduction of thermal requirements for recycling of the composite. Herein, we will highlight in detail the main differences between our work and the studies highlighted.

In Xuechang Zhou et al. work titled "Recyclable, weldable, mechanically durable, and programmable liquid metal-elastomer composites" they show a liquid metal elastomer composite that transforms from insulator to conductor through mechano-training and its ability to have photothermal effects and recyclability¹². A key difference in our work is the usage of nano-sized liquid metal particles that allow us to exploit its dielectric and photothermal properties simultaneously for enhanced performance of DEAs through co-stimulation. The micro-sized particles used in Zhou et al work would not be suitable for DEAs as the formation of conductive pathways when strained would lead to dielectric breakdown. It is only through the careful selection of nano-size ranges that the unique synergistic effect between electric field and NIR light can be used to enhance DEAs. This feature of liquid metal nanoparticles can be elaborated by the Young-Laplace equation that indicates that a higher external force is required to deform liquid metal nanoparticles compared to microparticles¹³. Furthermore, in terms of recycling, we highlight that by tuning the hydrogen bonds between the polymer chains using liquid metal nanoparticles, the temperature requirements for recycling can be lowered.

In Lv et al. work titled "Stimulus-driven liquid metal and liquid crystal network actuators for programmable soft robotics", NIR and electric field are used for actuation owing to the photothermal and electrically conductive properties of a liquid metal colloidal ink¹⁴. The key difference is that their electrically driven actuation is based on joule heating that generates a change in liquid crystalline ordering, instead of Maxwell mechanism (electrostatic pressure) that drives our actuators. . Co-stimulation of both stimuli was not demonstrated as well.

2. The authors have done a great job for the motivation of this work and presenting the results, however the literature review on the materials aspect is weak. If the novelty of this study is introducing a novel LM-polymer nanocomposite with enabling properties, then a brief background needs to be given to the readers. There are recent review articles on liquid metal nanocomposites, multifunctional liquid metal elastomer composites, and liquid metal materials for soft robotics that are highly related to the study in terms and can support certain claims in the manuscript.

Thank you for the reviewer's suggestion. In the introduction, we have added a short section that distinguishes the usage of liquid metal nanoparticles compared to microparticles. In addition, recent reviews and works (listed below) have been included to support the claims that have been made in the manuscript.

On Page 3 of Main manuscript:

Unlike liquid metal microparticles that are conductive under mechanical strains and rely on thermal effects to achieve electrothermal and photothermal actuation^{9,24,25}, LMNPs remain insulative^{26,27}, preventing dielectric breakdown to allow Maxwell pressures and thermal effects to be used simultaneously for co-stimulation. In addition, nanoscale liquid metal particles have higher photothermal conversion efficiencies²⁶.

Additional references

25 Majidi, C., Alizadeh, K., Ohm, Y., Silva, A. & Tavakoli, M. Liquid metal polymer composites: From printed stretchable circuits to soft actuators. *Flexible and Printed Electronics* 7, 013002 (2022).

26 Chiew, C., Morris, M. & Malakooti, M. H. Functional liquid metal nanoparticles: synthesis and applications. *Materials Advances* (2021).

29 Malakooti, M. H. et al. Liquid metal supercooling for low-temperature thermoelectric wearables. *Advanced functional materials* 29, 1906098 (2019).

37 Chiew, C. & Malakooti, M. H. A double inclusion model for liquid metal polymer composites. *Composites Science and Technology* 208, 108752 (2021).

40 Hu, Y. et al. Graphene oxide encapsulating liquid metal to toughen hydrogel. *Advanced Functional Materials* 31, 2106761 (2021).

3. Page 4, line 164-166: “Elastic modulus and mechanical toughness increased with LMNPs addition at 15 and 33 wt%, due to the effective load transfer between the elastomer matrix and the presence of gallium oxide shells which are stiffer (Fig. 2b) 35,36”. Can you clarify the meaning of effective load transfer and its relationship with stiffness? This increase is because of the solid gallium oxide shell and the effective elastic modulus can be even predicted using a double inclusion micromechanics model for liquid metal nanocomposites. Reference 35 on graphene (a solid filler) – TPU composites appears out of place unless the interface failure is being studied.

The effective load transfer is the result of the hydrogen bond interactions that formed between the carboxyl functional groups of the elastomer matrix and the nanoparticles. When the elastomer is being deformed, stress is being transferred effectively from the polymer chains to the liquid metal nanoparticle, resulting in enhancement in mechanical toughness¹⁵. The strong interfacial interactions have been studied through contact angle (Supplementary Fig. 6) and FTIR (Supplementary Fig. 8-9) measurements.

It should be clarified that the load transfer effect does not lead to an increase in stiffness of the nanocomposite. The increase in stiffness can be attributed to two factors. First, is the formation of bound rubber. This refers to the interphase region between the matrix and particles that is formed from the strong interfacial interactions between the two components^{16,17}. As the mobility of polymer chains within this region is restricted from these interactions, a stiffening effect is exhibited^{16,17}. While bound rubber has been mainly reported for solid fillers^{16,17}, similar formation of polymer coatings through physical interactions around liquid metal has been demonstrated^{18,19}.

Second, the stiffening behaviour is the result of the reduction of the inclusion diameter to hundreds of nanometres while the thickness of the gallium oxide shell remains to be a few nanometres^{20,21}. As a result, the effects of stress from the gallium oxide shell and filler-elastomer interactions are elevated²¹. The experimental data was evaluated against theoretical predictions that utilized a three-phase double inclusion (DI) model to predict the elastic modulus of the nanocomposites.²⁰ The model composed of two steps. First, the liquid core and the gallium oxide shell is homogenized and taken as one single phase. This single phase is subsequently homogenized with the matrix to achieve the overall properties of the nanocomposite. The governing principles are illustrated in supplementary notes 4. Compared to the DI model that is homogenized based on matrix modulus ($E_m = 1.87$), the experimental data is slightly larger than the prediction. This can be attributed to the formation of bound rubber surrounding the particles that have been reported to be significantly stiffer than the actual matrix. The modulus of overall matrix (composed of bound rubber and the pristine PULM0) was estimated based on the DI model and fitted to the experimental data. Thus, it is estimated that the modulus of the overall matrix at 3, 8, 15 vol% (corresponding to 15, 33 and 50 wt%) corresponds to 2.15, 2.40 and 2.50 MPa. The increasing modulus of bound rubber with LMNPs is related to the greater amount of physical interactions from strong carboxyl groups.

On Page 4-5 of Main manuscript:

With the addition of LMNPs at 15 and 33 wt%, the mechanical properties were further enhanced (Fig. 2b). Increase in elastic modulus with LMNPs is compared with theoretical predictions based on the double inclusion model³⁷, at which a slightly larger experimental modulus is found (Supplementary Fig. 13). This can be attributed to the formation of bound

rubber that surrounds LMNPs due to hydrogen bond interactions³⁸. At this interphase, the mobility of polymer chains is restricted, resulting in stiffening effects³⁹. Also, the increase in modulus is related to stress effects from gallium oxide shell and particle-elastomer interactions that are elevated as the inclusion diameter is reduced to hundreds of nanometres while the thickness of the gallium oxide shell remains at a few nanometers^{27,37}. Improvements to mechanical toughness can be related to effective load transfer from the polymer chains to LMNPs owing to interfacial interactions⁴⁰.

On Page 14-15 of Supporting Information

Supplementary Fig. 13. Elastic modulus as a function of liquid metal nanoparticle (LMNPs) concentration (vol. %). Dashed red line refers to predictions by the double inclusion (DI) model based on the modulus of the PULM0 matrix (E_m) homogenizing with various concentration of liquid metal. Dashed blue line refers to the fitting based on estimated modulus of the overall matrix (bound rubber and pristine PULM0) at various concentrations of liquid metal.

Supplementary Note 4.

The experimental data was evaluated against theoretical predictions that utilized a three-phase double inclusion (DI) model to predict the elastic modulus of the nanocomposites (Supplementary figure 13)¹⁰. The model composed of two steps. First, the liquid core and the gallium oxide shell is homogenized and taken as one single phase. This single phase is subsequently homogenized with the matrix to achieve the overall properties of the nanocomposite. Further details of the model can be found from the framework established by Chiew and Malakooti who implemented the three-phase DI model to predict the properties of liquid metal composites¹⁰.

Compared to the DI model that is homogenized based on matrix modulus ($E_m = 1.87$) with a range of concentration of liquid metal, the experimental data is slightly larger than the

prediction. This can be attributed to the formation of bound rubber surrounding the particles that have been reported to be stiffer than the actual pristine matrix. The modulus of the overall matrix (bound rubber and pristine PULM0) was estimated based on the DI model and fitted to the experimental data. Thus, similar modulus of the overall matrix at 3, 8, 15 vol% (corresponding to 15, 33 and 50 wt%) was found from bound rubber effects, corresponding to 2.05, 2.10 and 2.0 MPa. While the formation of bound rubber is within tens of nanometers, Brune et al have showed through direct measurements, that the shear modulus of bound rubber can be within orders of magnitude greater than the pristine matrix¹¹. Thus, it is reasonable that the overall matrix is stiffer than predicted as its modulus is a combined effect of both bound rubber and the matrix.

4. In Figure 3e, it looks like increasing the NIR light intensity to 0.6 W/cm² increases the actuation significantly, but it does not fully recover like the case for 0.2 W/cm². Wouldn't this cause an issue in recovering the original shape? This irreversible behavior is also observable for the crawling robot demo in Figure 4c. The delay keeps increasing but it does not fully recover like the actuation in the first half without NIR light.

Thank you for the reviewer question. For figure 3e (now Fig. 5c), co-stimulation at higher NIR light intensity does lead to an increased actuation drift. This is attributed to the slower relaxation time compared to the actuation frequency, causing the actuation to drift from its starting state. The slower relaxation time is the result of longer cooling times and larger viscoelastic behaviours that is exhibited at higher NIR light intensity. Similarly, this actuation drift is often seen in highly viscoelastic elastomers such as VHB acrylic films that is commonly used for DEAs²². However, this does not imply that the original shape cannot be recovered. To demonstrate this, we evaluated the stride performance of the crawler. To evaluate these strides, the rear leg was locked to prevent locomotion and minimize slippage effects (Supplementary Fig. 37). Two conditions of co-stimulation was further explored. First, NIR light was applied at the same frequency as the electric field. Second, the NIR light was applied continuously while the electric field was driven at a frequency. Based on Supplementary Fig. 38, NIR light applied continuously led to increased flattening and longer strides but was accompanied by larger actuation drifts. Similar effects were observed at higher NIR light intensities (Supplementary Fig. 39). These findings can be attributed to the higher generated temperatures that led to longer cooling times and larger viscoelastic behaviours. Despite the larger drifts, all crawlers eventually recovered their original shapes. Particularly, recovery can even be achieved when driven by electric field alone. This is seen from 40 to 100 s of Supplementary Fig. 35-36 at which actuation drifts are being reduced while being driven by electric field. The speed of recovery is highly dependent on temperatures generated within PULM15 at which shorter recovery times were observed when lower NIR light intensity was utilized or when NIR light was applied at 1 Hz.

On Page 9 of Main manuscript:

Stride performances of the crawler was first being evaluated by locking the rear leg to minimize slippage effects and to ensure that the crawler remained within the measurement range of displacement sensors (Supplementary Fig 37). Two conditions of co-stimulation was explored. First, NIR light was applied at the same frequency as the electric field. Second, the NIR light was applied continuously while the electric field was driven at a frequency. NIR light applied continuously led to increased flattening and longer strides but was accompanied by larger actuation drifts (Supplementary Fig. 38). Similar effects were observed when NIR light intensities were increased (Supplementary Fig. 39). These findings is attributed to the higher

generated temperatures at longer exposure times. Upon removal of NIR light, the crawlers could gradually recover their shape while being operated by electric field alone or when stimuli were removed. The speed of recovery is highly dependent on temperatures generated within the film at which shorter recovery times were observed at lower NIR light intensity or when NIR light was applied at a frequency. This highlights the versatility of utilizing NIR light exposure times to control the performances of DEMES crawler.

On Page 29 of Supporting Information

Supplementary Fig. 37. Stride performance of DEMES soft crawler driven by electric field. **a** Schematic of experimental setup to measure the change in vertical displacement (Δy) of strides. The rear leg was fixed to analyze the stride behavior without locomotion. **b** Δy of the crawler when driven by an electric field of $35 \text{ V } \mu\text{m}^{-1}$ at 1 Hz. **c** Schematic of experimental setup to measure the change in horizontal displacement (Δx) of strides. **d** Δx of the crawler when driven by an electric field of $35 \text{ V } \mu\text{m}^{-1}$ at 1 Hz.

On Page 30 of Supporting Information

Supplementary Fig. 38 Stride performance of PULM15 DEMES soft crawler when driven by electric field ($35 \text{ V } \mu\text{m}^{-1}$) alone and by co-stimulation of NIR light (0.2 W cm^{-2}) and electric field ($35 \text{ V } \mu\text{m}^{-1}$). Change in **a.** vertical displacement (Δy) and **b.** lateral displacement (Δx) when driven by electric field ($35 \text{ V } \mu\text{m}^{-1}$) at 1 Hz for 20 s, followed by co-stimulation of NIR light (0.2 W cm^{-2}) at 1 Hz and electric field ($35 \text{ V } \mu\text{m}^{-1}$) at 1 Hz for 20s. After which, NIR light was removed and the crawler was being cooled while being driven by an electric field ($35 \text{ V } \mu\text{m}^{-1}$) for 60 s before being turned off. Change in **c.** vertical displacement (Δy) and **d.** lateral displacement (Δx) when driven by electric field ($35 \text{ V } \mu\text{m}^{-1}$) at 1 Hz for 20 s, followed by co-stimulation of NIR light (0.2 W cm^{-2}) applied continuously and electric field ($35 \text{ V } \mu\text{m}^{-1}$) at 1 Hz for 20s. After which, NIR light was removed and the crawler was being cooled while being driven by an electric field ($35 \text{ V } \mu\text{m}^{-1}$) for 60 s before being turned off. Larger drifts can be attributed to higher temperatures generated by continuous application of NIR light.

Supplementary Fig. 39 Stride performance of PULM15 DEMES soft crawler when driven by electric field ($35 \text{ V } \mu\text{m}^{-1}$) alone and by co-stimulation of NIR light (0.4 W cm^{-2}) and electric field ($35 \text{ V } \mu\text{m}^{-1}$). Change in **a.** vertical displacement (Δy) and **b.** lateral displacement (Δx) when driven by electric field ($35 \text{ V } \mu\text{m}^{-1}$) at 1 Hz for 20 s, followed by co-stimulation of NIR light (0.4 W cm^{-2}) at 1 Hz and electric field ($35 \text{ V } \mu\text{m}^{-1}$) at 1 Hz for 20s. After which, NIR light was removed and the crawler was being cooled while being driven by an electric field ($35 \text{ V } \mu\text{m}^{-1}$) for 60 s before being turned off. Change in **c.** vertical displacement (Δy) and **d.** lateral displacement (Δx) when driven by electric field ($35 \text{ V } \mu\text{m}^{-1}$) at 1 Hz for 20 s, followed by co-stimulation of NIR light (0.4 W cm^{-2}) applied continuously and electric field ($35 \text{ V } \mu\text{m}^{-1}$) at 1 Hz for 20s. After which, NIR light was removed and the crawler was being cooled while being driven by an electric field ($35 \text{ V } \mu\text{m}^{-1}$) for 60 s before being turned off. Larger drifts can be attributed to higher temperatures generated by continuous application of NIR light.

5. Please provide details on how the bending angles are calculated and comment on how robust your approach is. Based on the schematic in the supplementary figure 22, the bending angle is highly dependent on the selection of the two lines and can lead to a significant variation in the bending angle. I'm also curious to see how this bilayer actuator can achieve a bending angle of 120-140 degrees.

To measure the bending angle, lines were drawn in tangent to the actuator tip (Fig. R1a). The bending angles were determined using an image software (ImageJ) that measured the angle between the two lines tangent at the actuator tip before and during actuation (Supplementary Fig. 31). We agree with the reviewer that based on the selection of the two lines, there can be variations in the bending angles measured. As shown in Fig. R1b, this variation depends on the tangent point that is selected along the actuator. The further the tangent point is selected from the tip at point 1, 2 and 3, the larger the deviation from 7, 12 and 16 ° respectively. Consistent measurements can be obtained by ensuring that the tangent point is fixed at the tip. From Supplementary Fig 31, a bending angle of 127 ° is achieved by the bilayer angle. To provide better guidance on the measurement, we have included more details in the method section. In addition, an actual example of the bending angle measurement is provided in Supplementary Fig 31.

Figure R1 Bending angle of bilayer actuator **a** Red dashed lines represent the tangent at the actuator tip and light blue lines are drawn normal to the actuator tip. **b** Tangent lines drawn at point 1, 2 and 3 along the actuator shown a deviation of 7, 12 and 16°.

On Page 3 of Supporting Information

The bending angles of the DEMES were determined as the angle between the tangent lines at the actuator tip of initial and deformed states. These angles were measured using an image analysis software (Image J).

Supplementary Fig. 31. Measurement of bending angle between the initial and deformed state. The dashed red lines indicate the tangent at the actuator tip. Dotted black lines are drawn normal to the actuator tip.

6. In DSC results (Figure 1f and Figure S8), there are additional peaks for LM filled PU. These peaks are around -27 degrees C in the heating cycle as well as peaks at around -50 and -65 degrees C in the cooling cycle. Are these peaks related to the melting/freezing of LMNPs? It could be like the supercooling effect recently reported for liquid metal composites. Please elaborate more on this point in the manuscript.

Yes, as mentioned by the reviewer, the additional peaks are related to the melting and freezing of LMNPs. To investigate the supercooling effect, DSC measurements was performed on bulk liquid metal at the same heating rate of 10 °C/min under a nitrogen atmosphere (Supplementary Fig. 12). Bulk liquid metal showed a distinct freezing point and melting point of -5.8 and 11.5 °C respectively. In contrast, LMNPs displayed peak freezing points of -63.8 and -48.7 °C and peak melting points of -41.3 and -30.7 °C. The multiple phase transition peaks observed arise from the polydisperse distribution of LMNPs, that can be observed from dynamic light scattering measurements (Fig. 1d). The suppression of freezing and melting points is the result of supercooling effects that is observed when bulk liquid metal is confined to nanodroplets. Particularly, this effect has been attributed to the phase separation between solid and liquid phases and the formation of their interface²³.

On Page 4 of Main manuscript:

When compared to the thermal transition of bulk LM (Supplementary Fig. 12), a suppression of peak freezing (-5.8 to -63.8 and -48.7 °C) and melting points (11.5 to -41.3 and -30.7 °C) was observed for LMNPs. This is attributed to supercooling effects that arise from the phase separation between solid and liquid phases and the formation of their interface²⁹

Supplementary Fig. 12 DSC curve of bulk liquid metal.

Minor suggestions:

1. The x-axis in Figure 4c is time (ms). Is the time measured in milliseconds or it's a typo? The speed in Movie S1 indicates it should be in seconds.

Thank you for pointing out this error. The required changes have been done for Fig. 4c (now Fig 6c).

Fig. 6c Δy and Δx of two-anchor crawling locomotion as a function of time. First, the crawler was driven by an electric field of $35 \text{ V } \mu\text{m}^{-1}$ at 1 Hz. After which, NIR light (0.4 W cm^{-2}) was

applied continuously while the crawler was driven by the same electric field. When NIR light was applied, the enhanced stride led to a larger distance travelled

2. Dashed lines in Figure 3h says “Theory”. This can be misleading unless there is a theoretical model to predict the actuation. You may consider rephrasing this.

Thank you for the suggestion. We have rephrase “theoretical” to “expected additive effect”.

On Page 8 of Main manuscript:

... bending angles were found to be larger at lower electric fields in comparison to the expected additive effect of individual DEMES and photothermal actuation (Fig. 5e and Supplementary Fig. 34).

Fig 5. e Representative bending angles of PULM15 DEMES individually driven by electric field and simultaneously driven by electric fields and NIR light intensity of 0.2 W cm^{-2} . Dotted line represents the expected additive effects based on the addition of bending angles when driven by electric field and NIR light individually.

On Page 27 of Supporting Information

Supplementary Fig. 34. Representative bending angles of PULM15 DEMES individually driven by electric field and simultaneously driven by electric fields and NIR light intensity of

a 0.4 W cm^{-2} and **b** 0.6 W cm^{-1} . Dotted line represents the expected additive effects based on the addition of bending angles when driven by electric field and NIR light individually.

3. Page 7, line 287: “Fig. 4d and Supplementary Movie 4” This must be Movie 3 because there were only three SI movies.

Thank you for pointing out this error. The required changes have been done.

On Page 9 of Main manuscript:

Enhanced flattening of the crawler under co-stimuli of NIR light and electric field offers shape adaptation capabilities at narrow gaps (Fig. 6d and Supplementary Movie 3).

References

- 1 Full, R. J. & Meijer, K. Metrics of natural muscle function. *Chapter 3*, 73-89 (2001).
- 2 Duduta, M., Hajiesmaili, E., Zhao, H., Wood, R. J. & Clarke, D. R. Realizing the potential of dielectric elastomer artificial muscles. *Proceedings of the National Academy of Sciences* **116**, 2476-2481 (2019).
- 3 Mojtabai, K. D. *et al.* Diels–Alder Augmented Epoxies with Plasmonic Nanoparticle Fillers for Efficient Photothermal Depolymerization. *ACS Applied Polymer Materials* **4**, 2703-2711 (2022).
- 4 Ghosh, S. K. & Pal, T. Interparticle coupling effect on the surface plasmon resonance of gold nanoparticles: from theory to applications. *Chemical reviews* **107**, 4797-4862 (2007).
- 5 Jeon, B. *et al.* Photothermal polymer nanocomposites of tungsten bronze nanorods with enhanced tensile elongation at low filler contents. *Polymers* **11**, 1740 (2019).
- 6 Guo, Z. *et al.* Galvanic replacement reaction for in situ fabrication of litchi-shaped heterogeneous liquid metal-Au nano-composite for radio-photothermal cancer therapy. *Bioactive materials* **6**, 602-612 (2021).
- 7 Gan, T., Shang, W., Handschuh-Wang, S. & Zhou, X. Light-induced shape morphing of liquid metal nanodroplets enabled by polydopamine coating. *Small* **15**, 1804838 (2019).
- 8 Biggs, J. *et al.* Electroactive polymers: developments of and perspectives for dielectric elastomers. *Angewandte Chemie International Edition* **52**, 9409-9421 (2013).
- 9 Romasanta, L. J., López-Manchado, M. A. & Verdejo, R. Increasing the performance of dielectric elastomer actuators: A review from the materials perspective. *Progress in Polymer Science* **51**, 188-211 (2015).
- 10 Wallin, T. J. *et al.* 3D printable tough silicone double networks. *Nature communications* **11**, 1-10 (2020).
- 11 Cao, P.-F. *et al.* Robust and elastic polymer membranes with tunable properties for gas separation. *ACS Applied Materials & Interfaces* **9**, 26483-26491 (2017).
- 12 Chen, G. *et al.* Recyclable, weldable, mechanically durable, and programmable liquid metal-elastomer composites. *Journal of Materials Chemistry A* **9**, 10953-10965 (2021).
- 13 Bark, H., Tan, M. W. M., Thangavel, G. & Lee, P. S. Deformable High Loading Liquid Metal Nanoparticles Composites for Thermal Energy Management. *Advanced Energy Materials* **11**, 2101387 (2021).
- 14 Lv, P. *et al.* Stimulus-driven liquid metal and liquid crystal network actuators for programmable soft robotics. *Materials Horizons* **8**, 2475-2484 (2021).
- 15 Hu, Y. *et al.* Graphene oxide encapsulating liquid metal to toughen hydrogel. *Advanced Functional Materials* **31**, 2106761 (2021).
- 16 Qu, M. *et al.* Nanoscale visualization and multiscale mechanical implications of bound rubber interphases in rubber–carbon black nanocomposites. *Soft Matter* **7**, 1066-1077 (2011).
- 17 Brune, P. F. *et al.* Direct measurement of rubber interphase stiffness. *Macromolecules* **49**, 4909-4922 (2016).
- 18 Guo, R., Sun, X., Yuan, B., Wang, H. & Liu, J. Magnetic liquid metal (Fe-EGaIn) based multifunctional electronics for remote self-healing materials, degradable electronics, and thermal transfer printing. *Advanced Science* **6**, 1901478 (2019).
- 19 Liao, M., Liao, H., Ye, J., Wan, P. & Zhang, L. Polyvinyl alcohol-stabilized liquid metal hydrogel for wearable transient epidermal sensors. *ACS applied materials & interfaces* **11**, 47358-47364 (2019).

- 20 Chiew, C. & Malakooti, M. H. A double inclusion model for liquid metal polymer composites. *Composites Science and Technology* **208**, 108752 (2021).
- 21 Pan, C. *et al.* A Liquid-Metal–Elastomer Nanocomposite for Stretchable Dielectric Materials. *Advanced Materials* **31**, 1900663 (2019).
- 22 Zhang, X., Wissler, M., Jaehne, B., Breonmann, R. & Kovacs, G. in *Smart Structures and Materials 2004: Electroactive Polymer Actuators and Devices (EAPAD)*. 78-86 (SPIE).
- 23 Malakooti, M. H. *et al.* Liquid metal supercooling for low-temperature thermoelectric wearables. *Advanced functional materials* **29**, 1906098 (2019).

Reviewers' Comments:

Reviewer #1:

Remarks to the Author:

The manuscript has been improved significantly and the authors have addressed my comments and concern well. The manuscript is now well organized and well written. I don't have further comments on the manuscript, and it is ready to be accepted.

Reviewer #2:

Remarks to the Author:

In the revised version of the manuscript, the authors addressed most of my comments. I believe that the manuscript can be published in Nature Communications.

Reviewer #3:

Remarks to the Author:

The authors have successfully addressed all my comments in the revised manuscript. The new data and revised text clearly explains the novelty of the work and allows readers to better understand the behavior of these liquid metal composites.